



# Convective Organization and 3D Structure of Tropical Cloud Systems deduced from Synergistic A-Train Observations and Machine Learning

Claudia J. Stubenrauch[1], Giulio Mandorli[1], Elisabeth Lemaitre[1]

[1]Laboratoire de Météorologie Dynamique / Institut Pierre-Simon Laplace, (LMD/IPSL), Sorbonne Université, Ecole Polytechnique, CNRS, Paris, France

*Correspondence to*: Claudia J. Stubenrauch (claudia.stubenrauch@lmd.ipsl.fr)

**Abstract.** We are building a 3D description of upper tropospheric (UT) cloud systems in order to study the relation between convection and cirrus anvils. For this purpose we used cloud data from the Atmospheric InfraRed Sounder and the Infrared
Atmospheric Sounding Inferometer and atmospheric and surface properties from the meteorological reanalyses ERA-Interim and machine learning techniques. The different artificial neural network models were trained on collocated radar – lidar data from the A-Train in order to add cloud top height, cloud vertical extent, cloud layering, as well as a rain intensity classification (no, light or heavy) to other variables describing the UT cloud systems. The rain intensity classification has an accuracy of about 65 to 70% and allows to build objects of strong precipitation, used to identify convective organization. This classification
is more efficient to detect large latent heating compared to cold cloud temperature. The cloud system concept allows a process-oriented evaluation of parameterizations in climate models. In agreement with earlier studies, we found that the rain intensity is maximum after the first development of anvils and that deeper convection leads to larger heavy rain areas and a larger detrainment. Finally we have shown the usefulness of our data to investigate tropical convective organization. A comparison of different tropical convective organization indices and proxies to define convective areas has revealed that all indices show
a similar annual cycle in convective organization, in phase with the one of convective core height, anvil vertical extent, and horizontal detrainment of the mesoscale convective systems and in opposite phase with the one of the ratio of thin cirrus over total anvil size. Differences can be understood by seasonal cycles of size and number of areas in phase for intense precipitation and opposite phase for cold clouds as proxies. The geographical patterns and magnitudes in radiative heating rate inter-annual changes with respect to one specific convective organization index ($I_{org}$) for the period 2008 to 2018 are similar for both
proxies, but slightly larger for rain intensity, and they are similar to the ones related to the El Niño Southern Oscillation. However, the time series of the inter-annual anomalies of convective organization depend on the convective organization index.



## 1 Introduction

Upper tropospheric (UT) clouds represent about 60% of the total cloud cover in the deep tropics (e. g. Stubenrauch et al., 2013, 2017). These clouds, when created as anvil outflow from deep convection, often build large systems (e.g. Houze, 2004). Observational and CRM studies (e.g. Del Genio and Kovari, 2002, Posselt et al., 2012) have shown that tropical storm systems over warmer water are denser with more intense precipitation and cover wider areas than those over cooler water. Thin cirrus surround the highest anvils (Protopapadaki et al., 2017), which may be explained by UT humidification originating from deep convection (e.g. Su et al., 2006). Their structure and amount may respond to changing convection induced by climate warming. Organized convection, leading to MCSs and therefore associated to extreme precipitation, is a research subject of high interest, in particular in regard to climate warming, and many results have been published (e. g. Popp and Bony, 2020; Bony et al., 2020, Pendergast et al., 2020; Blaeckberg and Singh, 2022).

To study the relation between cirrus anvils and convection, we couple horizontal and vertical structure of UT clouds, including precipitation and 3D radiative heating. As single datasets are incomplete, we use their synergy and machine learning to get a more complete 3D description as well as simultaneous information on precipitation. A cloud system approach makes it possible to link the anvil properties to convection. Furthermore, the horizontal structure of intense rain areas within these cloud systems can be used to derive tropical convective organization indices.

The cross-track scanning Atmospheric Infrared Sounder (AIRS) and the Infrared Atmospheric Sounding Inferometers (IASI), aboard the polar orbiting Aqua and Metop satellites, provide cloud properties (CIRS, Clouds from IR Sounders, Stubenrauch et al., 2017) with a large instantaneous horizontal coverage. These have been used to reconstruct UT cloud systems (Protopapadaki et al., 2017). The good spectral resolution of IR sounders makes them sensitive to cirrus, down to a visible optical depth of 0.1, during daytime and nighttime. The vertical cloud structure is derived by combined radar-lidar measurements of the CloudSat and CALIPSO missions (Stephens et al., 2018), but only along successive narrow nadir tracks separated by about 2500 km. In order to get a more complete instantaneous picture, required for process studies, Stubenrauch et al. (2021) have demonstrated that the radiative heating rate profiles derived along these nadir tracks (CloudSat FLXHR-lidar, Henderson et al., 2013) can be horizontally extended by neural network regression models applied on cloud properties retrieved from AIRS and atmospheric and surface properties from meteorological re-analyses from the European Centre for Medium-Range Weather Forecasts (ECMWF). The 15-year time series revealed a connection of the heating by mesoscale convective systems (MCSs) in the upper and middle troposphere and the (low-level) cloud cooling in the lower atmosphere in the cool regions, with a correlation coefficient equal to 0.72, consolidating the hypothesis of an energetic connection between the convective regions and the subsidence regions.

In this article we present additional variables expanded to the horizontal coverage of AIRS and IASI by machine learning models, trained with collocated CloudSat-lidar retrievals: cloud top height, cloud vertical extent, cloud layering (above and below the clouds identified by CIRS), as well as a precipitation intensity classification (no, light or heavy). Section 2 includes





a description of the collocated data, the neural network development as well as an evaluation of the predictions and the construction of the tropical horizontal fields of these additional variables. Furthermore we present in this section the cloud system reconstruction and metrics on convective organization derived from areas with heavy precipitation. Section 3 highlights

results which show the applicability of these newly derived variables: comparison of properties of precipitating and non-precipitating UT clouds, statistical analysis of the behaviour of MCS properties during their life cycle and in respect to their convective depth. Finally we explore the annual cycle of tropical convective organization and geographical patterns in changes of the radiative heating rate fields with respect to inter-annual changes in tropical convective organization. Conclusions and an outlook are given in Section 4.

**2 Data, Methods and Evaluation**

Satellite observations become a major tool to observe our planet. However, they do not provide instantaneous complete views, because passive remote sensing is not able to provide the vertical structure of clouds and active radar-lidar measurements are only available along very narrow nadir tracks. By training neural networks we combine the complementary information from passive and active remote sensing and build a more complete 3D structure of clouds.

**2.1 Collocated AIRS – CloudSat-lidar – ERA-Interim data**

The satellite observations used for the training originate from the A-Train constellation (Stephens et al., 2018), with local overpass times around 1:30 AM and 1:30 PM. As input variables we use cloud properties retrieved from AIRS measurements by the CIRS (Clouds from IR Sounders) algorithm (Stubenrauch et al., 2017) and coincident atmospheric and surface properties from meteorological reanalyses ERA-Interim (Dee et al., 2011). The target variables are products derived from combined radar

– lidar measurements from the CloudSat and CALIPSO missions. Cloud top height, cloud vertical extent (difference between cloud top and cloud base) and number of vertical cloud layers are given by the CloudSat 2B - GEOPROF – lidar dataset (Mace et al., 2009), while the precipitation rate and quality are given by the 2C – PRECIP-COLUMN dataset (Haynes et al., 2009). We collocated these datasets over the period 2007 to 2010, as described in Stubenrauch et al. (2021), and used the latitude band 30N – 30S for the training and application. Input and target variables are presented in Table 1.

Cloud types are defined according to cloud pressure ($p_{cld}$) and cloud emissivity ($\varepsilon_{cld}$) from AIRS-CIRS as: highlevel clouds with $p_{cld} < 440$ hPa, and further high opaque with $\varepsilon_{cld} > 0.95$, cirrus with $0.95 > \varepsilon_{cld} > 0.5$ and thin cirrus with $0.5 > \varepsilon_{cld} > 0.05$. Mid- and lowlevel clouds ($p_{cld} > 440$ hPa), and further midlevel ($p_{cld} < 680$ hPa) opaque with $\varepsilon_{cld} > 0.5$ and partly cloudy with $\varepsilon_{cld} < 0.5$, and lowlevel ($p_{cld} > 680$ hPa) opaque with $\varepsilon_{cld} > 0.5$ and partly cloudy with $\varepsilon_{cld} < 0.5$.




**Table 1: List of variables for the prediction of cloud vertical structure and precipitation rate.**

**Input**

*Clouds*

| | |
|---|---|
| CIRS cloud properties and uncertainties | $\varepsilon_{cld}$, $p_{cld}$, $T_{cld,}$ $z_{cld,}$ $d\varepsilon_{cld}$, $dp_{cld}$, $dT_{cld}$, $dz_{cld}$, $\chi_{min}^2$ |
| cloud spectral emissivity difference | ($\varepsilon_{cld}$ (12$\mu$m) - ($\varepsilon_{cld}$ (9$\mu$m)) |

*Atmosphere*

| | |
|---|---|
| AIRS $T_B$ at 0.5° x 0.5° | $T_B$(11.85$\mu$m), $\sigma(T_B)$, $T_B$(7.18 $\mu$m) |
| ERA-Interim atmospheric properties | total precipitable water, $p_{tropopause}$ |
| ERA-Interim relative humidity profile | RH (determined from T and water vapour) within 10 layers |

*Surface*

| | |
|---|---|
| ERA-Interim surface properties | $p_{surf}$, $T_{surf}$, nb of atm. layers down to $p_{surf}$ |
| IASI spectral surface emissivity | $\varepsilon_{surf}$(9, 10, 12$\mu$m)     (monthly mean climatology) |
| day-night flag, land-ocean flag | |

**Target / Output**

| | |
|---|---|
| Cloud top height | $z_{top}$ |
| Cloud vertical extent | DZ = $z_{top}$ - $z_{base}$ |
| Cloud layers below | 0 or 1 |
| Cloud layers above | 0 or 1 |
| Rain rate | 0: no rain, 1: rain rate < 5 mm/hr, 2: rain rate > 5 mm/hr |
| Certain rain | 0: no, possible or likely rain,     1: certain rain |

## 2.2 Artificial Neural Network predictions and evaluation

### 2.2.1 Development of prediction models

We developed artificial neural network (ANN) regression models for cloud top height and cloud vertical extent and classification models for cloud vertical layering, separately for high-level clouds and for mid- / low-level clouds, over ocean and over land.

The prediction of the rain rate is the most difficult, partly because its distribution is highly skewed with a very large peak at 0 mm/hr. Therefore we only predict a 'rain rate classification', with three classes: 0: no rain, 1: small rain rate (> 0 mm/hr and < 5 mm/hr) 2: large rain rate (> 5 mm/hr). The CloudSat 2C-PRECIP-COLUMN data also provide a quality flag, varying between no, possible, likely and certain rain. We transformed this flag into a binary flag with 1 for certain rain and 0 else. Due to the skewedness of the distributions, we introduced class weights for the training, to balance statistics, comparing (0.25, 0.25 and 0.5) and (0.2, 0.3, 0.5) for the rain rate classification, and (0.5, 0.5) and (0.4, 0.6) for the determination of certain rain. We also tested a model development separately for three cloud scenes (high opaque, cirrus / thin cirrus and mid- / lowlevel clouds) and for two cloud scenes (high clouds excluding thin cirrus and mid- / lowlevel clouds), over ocean and over land. The samples for the development of these scene type dependent models vary from 4.8 million data points for mid- and low-level clouds over ocean to 94000 data points for opaque high-level clouds over land.

For the regression models, the final ANNs consist of an input layer with the approximately 30 input variables (Table 1), one hidden layer with 64 neurons, one with 32 neurons, one with 16 neurons and one output layer. We used the rectified linear unit





(ReLU) layer activation function. For the classification models, we use another activation function for the output layer (Sigmoid for binary classification and Softmax for multi-classification). Furthermore, we use the Adaptive Moment Estimation (Adam) optimizer with a learning rate of 0.0001 and a batch size of 256. For the training, we use randomly chosen 80% of the dataset. The remaining 20% are used for validation. The random data choice is stratified by day-night and cloud type (section 2.1), in order to have similar statistics in these portions.

As many input variable distributions are not Gaussian, and to avoid outliers, we determined for each variable acceptable minimum and maximum values, adapted to each scene for which the models were trained: ocean or land, high clouds or mid- / low-level clouds. Then we normalized the input variables by subtracting the minimum value and then dividing by the difference between maximum and minimum. Before the application of the models, all input variables are first bounded between these minimum and maximum values.

The model parameters are fitted by minimizing a loss function, corresponding to the average of the squared differences (SME) for the regression, and corresponding to the cross entropy for the classification, between the predicted and the target value.

2.2.2 Evaluation using collocated data

For the evaluation we use as metrics the mean absolute error (MAE) between the predicted and observed target values for the regression and the accuracy for the classification. In order to avoid overfitting, we stop the fitting when the minimum loss does

not further improve during twenty iterations (epochs). The accuracy (ratio of correctly classified samples and overall number of samples) for unbalanced datasets provides an overoptimistic estimation of the classifier ability on the majority class, and therefore we present the Matthews correlation coefficient (MCC) in Table 3. MCC produces only a high score if the prediction obtains good results in all of the four confusion matrix categories (true positives, false negatives, true negatives, and false positives), proportionally to both, the size of positive elements and the size of negative elements in the dataset. As MCC ranges

from -1 to +1, with MCC = 0 meaning a random result, we use the normalized MCC, (MCC+1)/2, which better compares with accuracy, with 0.5 meaning a random result.

Tables 2 and 3 present the uncertainties given by the MAE for the regression models and the normalized MCC for the classification models, separately for different cloud types, over ocean and over land. In the case of DZ and the classifications, we compare results for two modeling strategies:

1) We first develop a regression model for the prediction of $z_{top}$. Then the predicted $z_{top}$ is used as an additional input variable for the prediction of DZ. Finally predicted $z_{top}$ and DZ are used as additional input variables for the classifications of cloud layering, rain rate and certain rain. For $z_{top}$, DZ and cloud layering the models have been separately developed over high and mid- / lowlevel clouds, while for rain rate and certain rain the training datasets for high clouds have been further divided into Cb and Ci / thin Ci.





2) We determine each variable independently and don't use predicted variables in the prediction of DZ, cloud layering, rain rate and certain rain. Instead, for the rain rate and certain rain classification we exclude thin cirrus and use slightly different class weights (see above) for balancing the training statistics. For the prediction of cloud layers below, we exclude low-level clouds.

The MAEs and normalized MCCs are very similar for both strategies. The uncertainty of the cloud top height is about 1 km
for high- and midlevel clouds (6% and 9%) and about 0.5 km for low-level clouds (20%). The quartiles indicated by the boxes in Figure S1 are about half of the MAEs. The uncertainty of DZ varies from 0.5 km (37%) for low-level clouds to 2.9 km (33%) for Cb. The quartiles of the relative differences between predicted and observed DZ are about 25 to 35%. Mean biases are small (a few meters). The normalized frequency distributions of observed and predicted $z_{top}$ in Figure 1 agree quite well for each of the cloud types (Cb, Ci, thin Ci and mid-/ low-level clouds). It is interesting to note that the features of slightly
higher clouds and more midlevel clouds over land than over ocean are also well obtained by the predictions. However, the $z_{top}$ distributions of the predicted values are slightly narrower than the ones of the observations. The normalized frequency distributions of observed and predicted DZ in Figure 1 agree also very well for Ci, thin cirrus and mid-/ low-level clouds, with decreasing DZ with decreasing cloud emissivity and cloud height. However, the bimodality for Cb, with a large peak around 15 km corresponding to the convective towers and a smaller peak around 6 km, probably corresponding to thick anvils, could
not be reproduced. By investigating further, Figure S2 shows that for those Cb for which a DZ < 10 km is predicted, there is no bias, but when a DZ > 10 km is predicted, corresponding to most of the convective towers, DZ is underestimated on average by about 1.5 km over ocean and by about 2 km over land.

**Table 2 MAE and relative MAE for the prediction of $z_{top}$ and DZ, over ocean and over land. For DZ, results are shown for predicted $z_{top}$ included and not included as input parameter.**

| ocean | Cb | Ci | thin Ci | lowlevel | midlevel |
|---|---|---|---|---|---|
| $z_{top}$ | 0.8 km<br>4.4 % | 1.1 km<br>6.5 % | 0.90 km<br>5.0 % | 0.5 km<br>18.9 % | 0.8 km<br>8.8 % |
| DZ | 2.9 km<br>31 % | 2.4 / 2.5km<br>38 % | 1.2 / 1.3 km<br>32 % | 0.5 / 0.6 km<br>36 % | 1.8 / 1.9 km<br>82 % |
| land | Cb | Ci | thin Ci | lowlevel | midlevel |
| $z_{top}$ | 0.9 km<br>5.1 % | 1.3 km<br>7.0 % | 1.0 km<br>5.4 % | 0.6 km<br>21.3 % | 0.9 km<br>21.3 % |
| DZ | 3.2 km<br>39 % | 2.6 / 2.8 km<br>43 % | 1.4 / 1.5 km<br>37 % | 0.7 / 0.8 km<br>47 % | 2.0 / 2.1 km<br>91 % |


The normalized MCCs for the classifications of certain rain, rain rate, and cloud layers additional to the ones identified by CIRS are about 0.7. Merely the prediction of rain from thin cirrus is close to random. This is because thin cirrus do not




precipitate, and detected rain can only be linked to the clouds underneath, for which the CIRS data do not have any information. Therefore we trained the second model only for Cb and Ci, assuming no rain for thin cirrus. With this assumption we miss about 2% of rainy areas beneath thin cirrus.

**Table 3 Normalized Matthews correlation coefficient for the prediction of rain rate (no, small, large), certain rain, cloud layer above and below, over ocean and over land. Two results are compared: the first include predicted $z_{top}$ and DZ as input parameters, the second does not. Instead, we used the hypotheses of no rain from thin Ci and no clouds underneath low-level clouds.**

| ocean | Cb | Ci | thin Ci | lowlevel | midlevel |
|---|---|---|---|---|---|
| **rain rate** | 0.65 / 0.64 | 0.69 / 0.70 | 0.55 / - | 0.62 / 0.62 | 0.68 / 0.67 |
| **certain rain** | 0.68 / 0.68 | 0.67 / 0.68 | 0.52 / - | 0.55/ 0.57 | 0.65 / 0.68 |
| **cloud layer above** | 0.64 / 0.67 | 0.71 / 0.72 | 0.68 / 0.69 | 0.69 / 0.71 | 0.67 / 0.68 |
| **cloud layer below** | 0.54 / 0.55 | 0.67 / 0.67 | 0.65 / 0.65 | 0.56 / - | 0.69 / 0.67 |
| **land** | **Cb** | **Ci** | **thin Ci** | **lowlevel** | **midlevel** |
| **rain rate** | 0.63 / 0.63 | 0.65 / 0.70 | 0.52 / - | 0.58 / 0.59 | 0.61 / 0.62 |
| **certain rain** | 0.66 / 0.67 | 0.64 / 0.70 | 0.50 / - | 0.57 / 0.51 | 0.64 / 0.60 |
| **cloud layer above** | 0.66 / 0.71 | 0.74 / 0.74 | 0.66 / 0.70 | 0.70 / 0.72 | 0.67 / 0.70 |
| **cloud layer below** | 0.53 / 0.52 | 0.65 / 0.65 | 0.64 / 0.64 | 0.50 / - | 0.66 / 0.69 |

## 2.3 Construction of tropical horizontal fields

The results in section 2.2.2 do not clearly show which of both models is performing better. For the prediction of DZ the inclusion of the predicted $z_{top}$ may lead to slightly better results, as the quartiles are slightly smaller (Figure S1). For further investigation we have applied both sets of ANN models to the whole AIRS-CIRS - ERA-Interim dataset over the period 2004 – 2018.

For the construction of the convective organization indices (section 2.5), we have also applied these models on IASI-CIRS – ERA-Interim data at local observation times of 9:30 AM and 9:30 PM. This is possible, because the models use input variables which are available in both datasets.

While these new variables have been obtained from machine learning per footprint, the final dataset has been gridded to 0.5° latitude x 0.5° longitude. The substructure has been kept by averaging over the most frequent scene type, distinguishing between highlevel clouds, mid- / lowlevel clouds, clear sky, and by keeping the fraction of coverage by Cb, Ci, thin Ci, mid- / lowlevel clouds and clear sky. To give an information on the rain intensity, we constructed a 'rain rate indicator' at footprint resolution by combining both, rain rate classification and rain quality binary classification, with values of 0 (0 & 0), 1 (0 & 1),



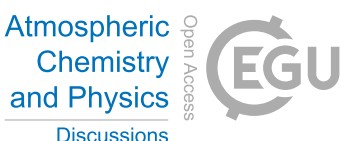

1.5 (1 & 0), 2.5 (1 & 1), 5 (2 & 0) and 7.5 (2 & 1). This rain rate indicator has then been averaged over 0.5°. In addition, we estimated the fractions within 0.5° of no rain and of certain rain as well as of light rain rate and of strong rain rate.

We illustrate the newly gained benefit by presenting in Figure 2 snapshots of the horizontal structure of some of these variables, at a specific day in January, once during a La Niña situation (2008) and once during an El Niño situation (2016), at two local times (1:30AM and 9:30PM). The gaps between orbits (corresponding to about 30% in the tropics) have been iteratively filled by the data closest in time. By using the data which are four hours apart, the data coverage has increased from 70% to 90%. Including also data which are 8 hours apart increases the coverage to 97%, and finally with those 12 hours apart leads to

complete coverage. These instantaneous horizontal structures, which are not possible to obtain from CloudSat-lidar data alone (Figure 1 of Stubenrauch et al., 2021), are quite different between La Niña and El Niño: While during the La Niña situation a very large multi-cell convective system evolved over Indonesia, the convective systems are more evenly distributed over the whole tropical band during the El Niño case. The latter can be explained by the shift of warmer SST towards the Central Pacific. The multi-cell convective cluster during the La Niña case shows bands of large DZ and rain rate, while during the El

Niño case these are more scattered. The different horizontal structure in precipitating areas over the tropical band between La Niña and El Niño suggests to derive a metrics for convective organization from these data (see Section 2.5). Figure 2 also indicates clouds above and below the CIRS clouds. We observe clouds below the edges of the cirrus anvils and multiple layer clouds in the region of thin cirrus bands. The latter are continued as very thin clouds above low-level clouds. All in all, these horizontal structures obtained from machine learning seem to be coherent, also those obtained from IASI, which are very

similar to those from AIRS.

When investigating monthly mean anomalies in the time series, we have seen a small artificial peak for the rain rate indicator in March 2014 for the AIRS observations. This peak was larger for the first model than for the second model. Therefore we show in the following all results using the second model which does not include predicted variables as input for the rain rate classification. At the end of this disturbance, most probably evoked by cosmic particles during a solar flare event, the AIRS

instrument shut down on 22 March, as its electronic circuit was affected. The instrument was operational again by end of March. No obvious failure is seen in the retrieved cloud variables, but many small areas with strong rain rate appear during this period.

**2.4 UT cloud system reconstruction**

The cloud system reconstruction is based on two independent variables, $p_{cld}$ and $\varepsilon_{cld}$, over grid cells of 0.5° latitude x 0.5°

longitude. This method is different with respect to other mesoscale cloud system analyses based on IR brightness temperature alone (e. g. Machado et al., 1998; Roca et al., 2014). After the filling of data gaps between adjacent orbits (Protopapadaki et al., 2017), UT cloud systems were built from adjacent elements of similar cloud height (within 50 hPa, $p_{cld}$ < 440 hPa). In a next step, the cloud emissivity distinguishes between convective cores ($\varepsilon_{cld}$ > 0.98), cirrus anvil (0.98 > $\varepsilon_{cld}$ > 0.5) and



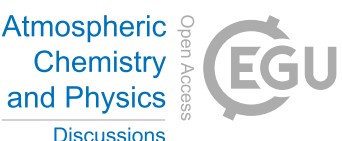

surrounding thin cirrus ($0.5 > \varepsilon_{cld} > 0.05$). In order to reduce the noise in the determination of the number of convective cores,

one searches for grid cells with $\varepsilon_{cld} > 0.98$ within regions of $\varepsilon_{cld} > 0.90$. The convective core fraction within a MCS is then the total number of these grid cells divided by the number of grid cells belonging to the whole system; and the number of convective cells corresponds to the number of regions with $\varepsilon_{cld} > 0.90$ which include at least one grid cell with $\varepsilon_{cld} > 0.98$. Each of these regions with at least one such grid cell counts as convective core. The original UT cloud system reconstruction was used to study the amount of surrounding thin cirrus as a function of convective depth. In order to maximize the coverage of

thin cirrus, a cell was declared to be covered by UT clouds, when it was already covered by 70% of UT clouds. These UT cloud systems cover about 25% of the latitude band between 30N and 30S.

Since these cloud systems reached very large sizes, in particular over the tropical Warm Pool, we revised the reconstruction by merging only grid cells which contain at least 90% UT clouds and of similar height defined within 6 hPa x $\ln(p_{cld}/hPa)$ (corresponding to 27 hPa for $p_{cld} = 100\,hPa$ and to 37 hPa for $p_{cld} = 400\,hPa$). In addition, the threshold for defining the regions

in which one looks for convective cores has been changed from 0.90 to 0.93. With this definition, the UT cloud system coverage is reduced to 20% within the latitude band 30N - 30S. MCSs with at least one convective core cover 15% of this latitude band, while the coverage of all UT clouds ($p_{cld} < 440\,hPa$) is about 35%.

Figure 3 compares the normalized frequency distributions of the cloud top and cloud base height as well as the normalized vertical extent, $DZ/z_{top}$, of the convective cores, cirrus anvils and surrounding thin cirrus within the MCSs for the 30% warmest

(SST > 302 K) and coolest (SST < 300 K) tropical ocean. First of all, the cloud top height distributions of all parts are very similar, with a slightly higher top of the convective cores. As expected, the cloud systems reach a much higher top in the warm regions than in the cool regions. For the other two variables, the distributions of convective cores and of thin cirrus are well separated, with the convective core base height in general smaller than 6 km and $DZ/z_{top}$ larger than 0.6 (filling more than 60% between surface and cloud top). The distribution of the cirrus anvils lies in between. We observe that the overlapping between

cirrus anvils and convective cores is larger for the cooler ocean regions. This indicates that the convective cores of the systems in these regions, which are less high than in the warmer, more convective regions, are probably less well defined by $\varepsilon_{cld} > 0.98$ than the ones of the MCSs in the warmer regions. Since we have now the normalized vertical extent from the machine learning, we can use it to improve the definition of convective cores, by adding the condition $DZ/z_{top} > 0.6$. All grid cells which do not fulfil the condition $DZ/z_{top} > 0.6$ are then counted back as cirrus anvil.

## 2.5 Indicators of tropical convective organization


Convective aggregation, which refers to the clustering of convective cells, occurs at multiple spatial scales in the tropics. Organized convection, leading to MCSs and therefore associated to extreme precipitation, is a research subject of high interest, in particular in regard to climate warming. The creation and maintenance of MCSs is strongly dependent on the available moisture in the lower troposphere and is influenced by wind shear. With the spatial resolution of our data we are mainly able





to consider the organization of MSCs into large squall lines, hurricanes or super-clusters. This type of organization should be more influenced by the large-scale environment and circulation.

There are two main factors that define the degree of organization for a given dataset: the variable used to define convection (section 2.5.1) and the metric used to compute the degree of organization (section 2.5.2).

2.5.1 Definition of convective areas within UT clouds

Studies have used cold IR brightness temperatures (e. g. Tobin et al., 2012; Bony et al., 2020) as well as precipitation rate (e. g. Popp and Bony, 2020; Blaeckberg and Singh, 2022) to define convective objects for the determination of convective organization metrics.

In order to estimate the organization of convection, measures of convection without missing data are needed. Since both AIRS and IASI data still show gaps of missing data between the orbits, we have filled these gaps with the measurements that are
nearest in time. First we excluded snapshots which have a data coverage in the latitudinal band 30N − 30S less than 68% for AIRS and less than 74% for IASI (as the swath is slightly larger for IASI). This ensures complete orbits. Gaps between orbits are then iteratively filled by using the observations closest in time. In general with four observations per day we get complete snapshots (coverage larger than 99.5%).

In general, strong vertical updraft, strong precipitation and very cold and optically thick cloud tops indicate deep convective
towers. Cold and optically thick cloud tops can be identified by a threshold in IR brightness temperature, TB, a measurement available by any radiometer aboard geostationary and polar orbiting satellites over a long time period. However, as this variable depends on both cloud height and emissivity (Figure 2 of Protopapadaki et al., 2017), for TB > 230 K, very cold semi-transparent cirrus may be misidentified as lower opaque clouds, leading to uncertainties in the sizes of the convective areas.

Figure 4 compares latent heating (LH) profiles derived from the precipitation radar measurements of the Tropical Rain
Measurement Mission (TRMM) for the same percentile statistics, using the coldest TB, the largest precipitation intensity (given by a large ML deduced rain indicator) and the largest horizontal extent of rain within each grid cell of 0.5° (given by the fraction of any precipitation deduced by ML). These LH profiles have been retrieved by the Spectral Latent Heating (SLH) algorithm (Shige et al., 2009) and are averaged over 0.5°. The time interval with the AIRS-CIRS data is within 20 minutes. The same percentile statistics allows to directly compare the efficiency of each variable to identify large latent heating, an
indicator of deep convection. In all cases the LH increases with decreasing TB, increasing rain rate indicator and increasing horizontal rain coverage per grid cell, showing that all variables can be used as proxies for deep convection. Moreover, at fixed percentiles the ML derived rain rate indicator as well as the grid cell rain coverage both lead to a larger LH than TB. This means that the ML derived rain rate classification, together with the CIRS identification of UT cloud, is a slightly better proxy for regions of large latent heating than TB.




### 2.5.2 Convective organization indices

It is not easy to define suitable organization metrics. The organization index $I_{org}$ (e. g. Tompkins and Semie, 2017) compares a cumulative distribution of nearest-neighbour distance (NNCDF) to one of the expected by randomly distributed points in the domain. $I_{org}$ lies between 0 and 1, with 0.5 corresponding to randomly distributed objects, and $I_{org} > 0.5$ indicates an organized
state. However, Weger et al. (1992), who initially developed this method to study the distribution of cumulus clouds, pointed out that the NNCDF is sensitive to the number of areas and to their size, in particular when the total area is larger than 5 to 15% of the studied domain: In the latter case, possible merging of the objects leads to a decrease of $I_{org}$. When using $I_{org}$, one has therefore to use a proxy for the definition of convective areas which corresponds to a total area that only covers a small fraction of the area to be studied.

Therefore White et al. (2018) developed the convective organization potential (COP), by assuming that 2D objects that are larger and closer together are more likely to interact with each other in the horizontal plane. It uses the distance between the centers of the objects and radii of equal area circles. Jin et al. (2022) have further developed COP to the area-based convective organization potential (ABCOP) by using the area rather than the radius and by changing the distances between centers to distances between outer boundaries. Furthermore the interaction potentials are computed for only one pair per aggregate and
summed up instead of averaged over all pairs. It is however very sensitive to the total area of the objects (section 3.3).
The Radar Organization MEtric (ROME) developed by Retsch et al. (2020) considers the average size, proximity and size distribution of the convective objects in a domain and is similar to COP, but like ABCOP it employs the distance between the outer boundaries. ROME defines interactions between pairs by assigning a weight to each pair that decreases with the distance and increases essentially with the area of the larger area, adding a contribution of the smaller area, depending on the separation
distance. It is given in units of $km^2$ and lies between the mean area of the objects and twice their mean area. Hence ROME is very sensitive to the mean areas of the objects (section 3.3).

## 3 Results

In this section we highlight results on the ML derived variables by investigating relationships. The cloud system approach enables us to study the behaviour of these variables within MCSs with respect to their life cycle stage and convective depth.
In particular, the rain rate classification and its horizontal and temporal expansion allow us to derive indices of tropical convective organization and to investigate the difference in UT cloud properties and their environment between periods of small and large tropical convective organization.

### 3.1 Properties of UT clouds

First we test the coherence between the ML derived rain rate classification and the collocated TRMM LH profiles already
presented in section 2.5.1. Figure 5 compares the LH profiles averaged over all UT clouds and over those with no rain, light



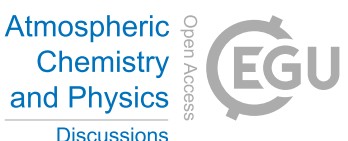

rain and heavy rain according to the rain rate classification described in section 2.3. Indeed, when the rain rate classification indicates no rain, the latent heating from TRMM is very small. The latent heating is on average about ten (five) times larger for grid cells which include heavy precipitation than the tropical average for UT clouds (mid- and lowlevel clouds). While latent heating profiles have a peak between 400 and 500 hPa for heavily precipitating UT clouds, the peak lies around 850 hPa

for strongly precipitating mid- and lowlevel clouds. This shows that the ML derived rain rate classification seems to be coherent for UT clouds as well as for lower clouds, though the noise for the latter may be larger.

Figure 6 compares normalized distributions of $\varepsilon_{cld}$, $z_{top}$, 'cloud fuzziness', $(z_{top} - z_{cld})/DZ$, and $DZ/z_{top}$ of precipitating and non-precipitating UT clouds. Cloud fuzziness increases with the difference between cloud top height and the cloud height retrieved by CIRS. The latter corresponds to the height at which the cloud reaches an optical depth of about 0.5 (Stubenrauch et al.,

2017). From these figures we clearly deduce that precipitating UT clouds in the tropics have an emissivity close to 1 and are in general higher, have a less fuzzy cloud top and have a larger vertical extent than non- precipitating UT clouds. These results are coherent with expectations and again confirm the quality of the rain rate classification derived by our machine learning procedure.

**3.2 Process-oriented behaviour of mesoscale convective systems**

The cloud system concept described in section 2.4 allows to link the convective core and anvil properties and therefore a process-oriented evaluation of parameterizations in climate models (Stubenrauch et al., 2019): The fraction of the convective core area within a cloud system indicates the life cycle stage (e. g. Machado et al., 1998), with a large fraction in the developing stage and a decreasing fraction during dissipation. Once the systems have reached maturity, the minimum temperature within

a convective core is a proxy for the convective depth.

3.2.1 Behaviour of precipitating areas

According to Takahashi et al. (2021), using a convection-tracking analysis on data from Intergrated Multisatellite Retrievals for GPM (IMERG), the fraction of precipitating cores (adjacent grid cells with a rain rate > 5 mm/hr) within precipitation systems (adjacent grid cells with rain rate > 0.5 mm/hr) first increases and then decreases during the evolution of these systems.

The maximum of the strong rain area relative to the whole precipitating area as well as the maximum and average intensity of the precipitation increase with the life time of the systems. This behaviour was also found by Roca et al. (2017).

Our data do not provide the absolute system life time, but the convective core fraction within a system indicates the maturity stage in a normalized life cycle. We also know that the coldest systems have also a tendency to live the longest (e. g. Rossow and Pearl, 2007; Takahashi et al., 2021). Figure 7 presents the evolution during the life cycle of a) the strong rain area relative

to the precipitating area within single core MCSs and b) the precipitating area relative to the whole MCS area. As the rain rate classification was obtained per CIRS footprint, a grid cell of 0.5° x 0.5° can be declared as precipitating by using different thresholds on the fraction with rain rate > 0 mm/hr. The same applies for grid cells including strong rain. Results using three different thresholds to define the precipitating and strongly precipitating areas are compared. For all thresholds, the systems





are precipitating already in the developing stage, while strong rain develops slightly later with a maximum just before reaching

maturity, in agreement with Fiolleau and Roca (2013) and Takahashi et al. (2021). During dissipation, relative rain area and
strong rain area decrease. The maximum rain area relative to the system area is about 0.45 when only grid cells are considered
which are completely covered by rain. The maximum strong rain area relative to the precipitating area within the MCSs is
about 0.12, when one considers a strong rain grid cell coverage of at least 0.2. It drops to 0.05 for a threshold of 0.5. To show
the behaviour of precipitating areas with respect to convective depth, Figure 7 also presents for mature MCSs (convective core

fraction within the system between 0.1 and 0.4, according to Protopapadaki et al., 2017), as a function of the minimum
temperature within the convective cores, c) the strong rain area relative to the precipitating area and d) the precipitating area
relative to the whole MCS area. Again the behaviour is the same for all three thresholds: deeper convection clearly leads to
larger areas of heavy rain within the precipitating areas in agreement with earlier studies, while the precipitating areas within
the system areas decrease, because the anvils are much larger in these deep convective cloud systems than in less deep

convective systems (see next section).

### 3.2.2 Behaviour of convective core and anvil properties

Figure 8 presents the evolution during the life cycle of convective core and anvil properties: a) core size, b) core top height, c)
vertical extent of the thick anvil and d) emissivity of the thick anvil, separately for maritime and continental MCSs. The
behaviour is similar, with increasing core size and core top height during the development phase and a decrease in the

dissipating stage. Oceanic systems reach slightly larger core sizes and core top heights than continental systems. However, the
core size depends on the definition of convective core. Our definition may lead to a larger size than a definition based on strong
vertical updraft. While the thick anvil emissivity decreases nearly linearly and has equal values for MCSs over land and over
ocean, the thick anvil vertical extent reaches a maximum only after the anvil developed, with a larger extent over ocean than
over land in the same development stage. Only towards dissipation the vertical extents are similar.

According to Figure 9, properties of mature MCSs (with a convective core fraction between 0.1 and 0.4) differ with convective
depth: the relative area of thin cirrus and the ratio of thick anvil volume over convective core volume, the latter gives an
indication of detrainment, both increase with convective depth, with similar values over ocean and over land. It is interesting
to note that the thick anvils of MCSs with larger convective depth have a slightly smaller emissivity but larger vertical extent
than those with smaller convective depth. The latter leads to a larger thick anvil volume, while the first indicates a smaller ice

water content.

Finally we compare in Figure 10 the MCS properties of those over oceanic regions with the 30% warmest SSTs and those with
the 30% coolest SSTs. The warmest oceanic regions in the tropics are the ones where deep convection, given by large core top
height, develops, with a larger fraction of strong rain, deeper and larger anvils, but also with a smaller anvil emissivity and
more surrounding thin cirrus. The latter have an effect on the radiative heating.





### 3.3 Tropical convective organization

In this section we explore and compare different metrics of convective organization, using different proxies for convection described in section 2.5. A spatial resolution of 0.5° relates to an organization of MSCs at a scale which is more linked to the large-scale environment and circulation. We first consider the annual cycle of convective organization and then highlight an application on inter-annual variability.

Figure 11 presents the annual cycle of $I_{org}$, ROME and COP, of the total and mean sizes of the convective areas and their number over the tropics. We compare results using different variables to define these areas, in particular precipitation intensity (given by ML derived rain rate indicator) of UT clouds ($p_{cld} < 350$ hPa) and cold cloud temperature ($T_{cld} < 230$ K) of opaque clouds ($\varepsilon_{cld} > 0.95$). The latter definition is similar to TB < 230 K, but without any contamination of colder thinner Ci. Since ROME is strongly related to the size of the convective areas, we have also computed the annual anomalies of the three indices by using a constant 2% of areas with the largest precipitation intensity. A similar approach was undertaken in a study by Blaeckberg and Singh (2022), using the precipitation intensity and ROME as proxies for convection and convective organization, respectively.

All three indices reveal a clear annual cycle of convective organization, with a minimum in April and November and a maximum in July / August and in January / February, though with differences in magnitude and width of the oscillations due to the choice of proxy for convective area. In Figure 12 we observe a seasonal behaviour in phase with the one of convective organization for the core height of the MCSs, anvil vertical extent (Figure S3), and anvil horizontal detrainment, the latter estimated by the ratio of anvil over convective core size. The opposite annual cycle of the fraction of single core MCSs confirms that convective organization corresponds to multi-core MCSs. In that case the ratio of thin cirrus over total anvil size is smaller, which may be explained by the fact that clustering of convective systems leaves less space between them for thin cirrus. The average tropical UT humidity is slightly higher between May and September, this is also the period where the surface temperature underneath the MCSs is largest (Figure S3). It is interesting to note that the minima and maxima of the relative subsidence area (clear sky and low-level clouds) are in accordance with the ones of convective organization. This then leads to a relation between the upper and middle tropospheric heating by the MCS and the cooling of the lower troposphere in the cooler subsidence regions, which has recently been found by Stubenrauch et al. (2021).

By comparing the choice of proxies for the definition of convective areas, we observe that the seasonal anomalies of COP are the less sensitive, while those of ROME are the most sensitive. The latter look very similar to those of the mean size of the convective areas (with a correlation coefficient larger than 0.9), and the minima and maxima in the cycle of intensive rain horizontal extent are shifted compared to those of cold opaque cloud horizontal extent. Thus the seasonal anomalies of ROME primarily reflect the ones of the mean areas of convection. When using a fixed area of intense precipitation, the magnitude is much smaller and the shift between the maxima due to the choice of cold clouds or intensive rain disappears.



The absolute values of the indices, presented in Figure S4 of the supplement, depend more strongly on the proxy used to define convective areas, but this dependency varies: $I_{org}$, reflecting the distance between convective areas, is larger over the whole annual cycle when one considers precipitation intensity instead of cold cloud temperature. The absolute maximum of COP is the same for all proxies during boreal summer, while for other seasons COP, like $I_{org}$, is larger when considering precipitation
intensity. ABCOP reflects the total tropical area, which seems to be more constant over the seasons when cold clouds are considered than when strongly precipitating clouds are selected. For the latter there are pronounced maxima in January and July. Note that the relative flatness of the seasonal cycle of the total tropical area of cold clouds can be explained by an opposite seasonal cycle in mean size and number of areas, while for intense precipitation their cycles are in phase.

This difference in seasonal behaviour indicates that the choice of the proxy for the definition of convective areas and of the
convective organization metrics plays a non-negligible role in the interpretation of results. The seasonal geographical distributions of intense precipitation occurrence (rain rate indicator > 2) in Figure S5 show that the regions of maximal occurrence also vary with season, in agreement with earlier results (e. g. Berry and Reeder, 2014). These may explain the difference in the absolute peak values of convective organization anomaly between boreal summer and boreal winter.

Of all the studied indices, ROME and ABCOP have a very high correlation with the mean area and the total area of convective
objects, respectively (larger than 0.8). However, these correlations depend on the domain and on the domain's spatial resolution. For the specific domain (30N-30S) and spatial resolution (0.5°) of this analysis, $I_{org}$ is the metric that is less related to these variables, and consequently, it is also the one that contains the additional information on organization. Therefore, we use $I_{org}$ in the following to explore changes of different variables with respect to inter-annual convective organization anomalies.

Changes in gradients of tropospheric radiative heating relate to changes in atmospheric circulation. We use the 3D radiative heating rate (HR) fields described in Stubenrauch et al. (2021) and $I_{org}$ computed from convective areas defined once by strong precipitation and once by cold cloud temperature over the period 2008 to 2018. In order to remove the seasonal dependency, we computed the 121 12-month running mean anomalies for these variables. The geographical distribution of radiative heating change with respect to convective organization change are presented in Figure 13, separately for the upper (100-200 hPa), mid
(200-600 hPa) and low (600-900 hPa) troposphere and for the two proxies to define convective areas. These geographical maps have been obtained by linear regression per grid cell of the 121 pairs of heating rate and $I_{org}$ anomaly (see examples in Figure S6 in the supplement).

The geographical patterns and magnitudes in HR change with respect to change in $I_{org}$ are similar for both proxies, but with slightly larger derivatives for strong precipitation areas. This may be expected as intense precipitation should be a more direct
proxy for convection than cold cloud top. In general the derivatives are large because inter-annual changes in $I_{org}$ are very small, as shown in Figure 14. In the upper troposphere we observe increased heating North and South of the equator in the Central Pacific and a decrease over the West Pacific, while in the mid and low troposphere there is an increase in heating



around the equator over the whole Pacific and Indian ocean, and a decrease in heating over the Warm Pool and in the Atlantic. The HR pattern changes in the convective regions are induced by relative changes of thin cirrus, cirrus and high opaque clouds,

which are similar but not identical to the ones related to the El Niño Southern Oscillation (ENSO) during this period (Figure S7), with increasing convection close to the equator and increasing cirrus and thin cirrus around the equator. Indeed, the correlation between Iorg and the oceanic Niño index (ONI) is positive (with correlation coefficients of 0.7 and 0.3 for cold cloud temperature and precipitation intensity as proxy, respectively). In the stratocumulus regions off the Western coasts of the Americas and of Australia there seems to be less cooling in the low troposphere, probably due to a recent reduction in low-

level clouds, in particular in the NE Pacific, which was found in coincidence with a shift in the phase of the Pacific decadal Oscillation (Loeb et al., 2018, Loeb et al., 2020, Sun et al., 2022). The similarity between the maps obtained with the two selections validates once again the reliability of the rain rate indicator obtained with ML, and the slightly stronger patterns lead to the conclusion that strong precipitation is a slightly better proxy to define convective areas than cold temperature.

While the patterns of the derivatives of heating / cooling with respect to $I_{org}$ show a coherent picture, correlations between

deseasonalized inter-annual anomalies of $I_{org}$ (Figure 14) and of the tropical means of different variables like surface temperature, thin cirrus area and subsidence area are much noisier, because inter-annual changes of the tropical means are small. The correlations depend on the proxies for the definition of the convective areas and in particular on the metrics for convective organization. Already the time series of the inter-annual anomalies of the different indices have a different behaviour as can be seen in Figure S8 in the supplement. We have also investigated tighter thresholds on the variables which

define deep convection (like rain rate indicator > 2.5 or $T_{cld}$ < 210 K), however we are left with only about 0.5% total area, which increases the noise level. In addition, we found that the results also change when we exclude objects with the size of only one grid cell (not shown), as already pointed out by Jin et al. (2022).

## 4. Conclusions and Outlook

We have presented a methodology to extend spatially and temporally information on the cloud vertical structure and precipitation derived from active lidar and radar measurements of CALIPSO and CloudSat missions. This new approach made use of CIRS data obtained from advanced IR sounder measurements of AIRS and IASI combined with ERA-Interim reanalyses and machine learning technologies using ANN. This synergy brought us one step further to build a complete 3D dataset of UT cloud systems, at 0.5° spatial resolution, which can be used for process studies in order to better understand the relationship

between convection and resulting anvils and how they are impacted by and feed back to climate change.

Uncertainties, expressed as maximum absolute error, of cloud top height and cloud vertical extent predicted by ANN regression models with about 30 input variables are about 1 km (0.5 km) and 30% (40%) for UT (low-level) clouds, respectively. Classifications of cloud vertical layering (separately clouds above and below) and of precipitation rate (no, light and heavy rain) have an accuracy, given by the normalized Matthews correlation coefficient, of about 65 to 70%, except for rain rate of



low-level clouds for which the accuracy is only about 60%. Nevertheless, this method allows to study horizontal structures of these variables on specific snapshots in time. In order to have a complete instantaneous coverage, the remaining 30% due to gaps between the orbits have been filled iteratively with the four observations per day of AIRS and IASI data, starting with those closest in time (already leading to 90%). These snapshots have been used to compute indices of tropical convective organization. We compared in particular two proxies to define the convective areas: precipitation intensity and cold opaque

cloud. We could show that the newly developed precipitation intensity classification is slightly more efficient to detect large latent heating and therefore deep convection compared to the cold cloud temperature.

The cloud system approach developed by Protopapadaki et al. (2017) has been slightly modified, leading to smaller cloud systems with less surrounding thin cirrus, as the grid cells belonging to the systems need now to include at least 90% UT clouds (instead of 65%). The normalized vertical extent obtained from the ML approach was then employed to slightly improve

the identification of the convective cores, in particular in the cooler tropical regions. MCSs with at least one convective core cover about 15% of the latitude band 30N to 30S, while the coverage of all UT clouds is about 35%.

The cloud system concept allows to link the convective core and anvil properties and therefore a process-oriented evaluation of parameterizations in climate models (Stubenrauch et al., 2019). In agreement with earlier studies (e. g. Schumacher and Houze, 2003; Roca et al., 2014, Takahashi et al., 2021), we found that the rain intensity is maximum after the first development

of anvils and that deeper convection leads to larger heavy rain areas. These results also confirm the quality of the ML derived precipitation rate classification.

With increasing convective depth mature MCSs show an increase in detrainment, given by the ratio of thick anvil volume over convective core volume. While the vertical extent of the thick anvil increases, the average emissivity slightly decreases with convective depth. Deep convection develops in the warmest oceanic regions of the tropics, producing a larger fraction of strong

rain, deeper and larger anvils, but also have anvils with a slightly smaller average emissivity and more surrounding thin cirrus. The latter have an effect on the radiative heating.

Finally we have shown the usefulness of our data to investigate tropical convective organization. By comparing different organization metrics and proxies to define convective areas we have shown that the three indices $I_{org}$, COP and ROME indicate a similar annual cycle of convective organization. However, ROME is strongly correlated to the mean area of the objects.

Considering the annual cycle of size and numbers of the object areas as well as their total area over the tropics, we observe a pronounced seasonal cycle of the total area of intense precipitation, while the total area of cold clouds in the tropics stays stable over the annual cycle, linked to a seasonal cycle in phase of size and number in the first case and an opposite seasonal cycle in the latter. This interesting difference in the behaviour of the two proxies should be further explored in the future. The seasonal cycle of core height of the MCSs, anvil vertical extent and anvil horizontal detrainment is in phase with the one of

convective organization, while an opposite cycle of the fraction of single core MCSs confirms that convective organization is strongly linked to multi-core MCSs. In that case the ratio of thin cirrus over total anvil size is smaller, which may be explained



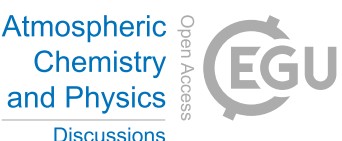

by the fact that clustering of convective systems leaves less space between them for thin cirrus. It is also interesting to note that the minima and maxima of the relative subsidence area are in accordance with the ones of convective organization.

Changes in gradients of tropospheric radiative heating relate to changes in atmospheric circulation. The geographical patterns and magnitudes in radiative heating rate inter-annual changes with respect to $I_{org}$ are similar for both proxies, but slightly larger for strong precipitation areas. This may be expected as intense precipitation should be a more direct proxy for convection than cold cloud top. The HR pattern changes are similar to the ones related to ENSO during this period.

However, the time series of the inter-annual anomalies of convective organization depend on the convective organization metrics, and correlations between these anomalies and those of the tropical means of different atmospheric variables do not
show consistent results. Therefore one has to be careful using only one of these organization indices and proxies to study climate change.

This data base of UT cloud systems, their vertical structure and precipitation areas is being constructed within the framework of the GEWEX (Global Energy and Water Exchanges) Process Evaluation Study on Upper Tropospheric Clouds and Convection (GEWEX UTCC PROES, https://gewex-utcc-proes.aeris-data.fr/) to advance our knowledge on the climate
feedbacks of UT clouds. In general, climate feedback studies are undertaken by climate model simulations, which rely upon their representation of convection and detrainment. We are now expanding the latent heating rates from TRMM onto this data base, using similar machine learning techniques. Once the total 3D diabatic heating is available, it will be used together with the information of the UT cloud systems to quantify the dynamical response of the climate system to the atmospheric heating induced by the anvil cirrus, refining and extending the studies of Li et al. (2013).

At present we are preparing a new version of CIRS data, using ERA5 (Hersbach et al., 2020) instead of ERA-Interim ancillary data in order to have a continuous dataset from 2003 to present, and newly calibrated AIRS L1C radiances (Manning et al., 2019) as input.

**Acknowledgements**
This work was supported by the Centre National de la Recherche Scientifique (CNRS), the Centre National d'Etudes Spatiales (CNES) and the TTL-Xing ANR-17-CE01-0015 project. The authors thank the members of the AIRS, CALIPSO, CloudSat, IASI and TRMM science teams for their efforts and cooperation in providing the data, as well as the engineers and space agencies who control the data quality. AIRS CIRS and IASI CIRS data have been produced by the French Data Centre AERIS.

**Code/Data availability**
All satellite L2 data used are publicly available and have been downloaded from their official websites. CIRS L2 data are distributed at https://cirs.aeris-data.fr (last access: 15 January 2021, CIRS, 2021). The TRMM latent heating rates correspond to Tropical Rainfall Measuring Mission (TRMM) (2018), GPM PR on TRMM Spectral Latent Heating Profiles L3 1 Day



0.5x0.5 degree V06, Greenbelt, MD, Goddard Earth Sciences Data and Information Services Center (GES DISC),
doi:10.5067/GPM/PR/TRMM/SLH/3A-DAY/06 (last access 15 May 2020). Monthly indices of the oceanic Niño index (ONI) were obtained from NOAA (https://origin.cpc.ncep.noaa.gov/products/analysis_monitoring/ensostuff/detrend.nino34.ascii .txt, last access: February 2020, NOAA-ONI, 2020). The ERA-Interim reanalysis dataset was downloaded from the Copernicus Climate Data Store. The CloudSat-lidar data have been provided by the AERIS ICARE data and services center (https://www.icare.univ-lille.fr/).


**Author contribution**

CJS developed the concept, improved the ANN method, analysed the cloud system related analysis and wrote the manuscript. GM produced and analysed the longterm dataset, computed and analysed the convective organization metrics and contributed to improvements of the manuscript. EL helped to develop the ANN models.


**Competing interests**

The authors declare no competing interests.

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





# Figures

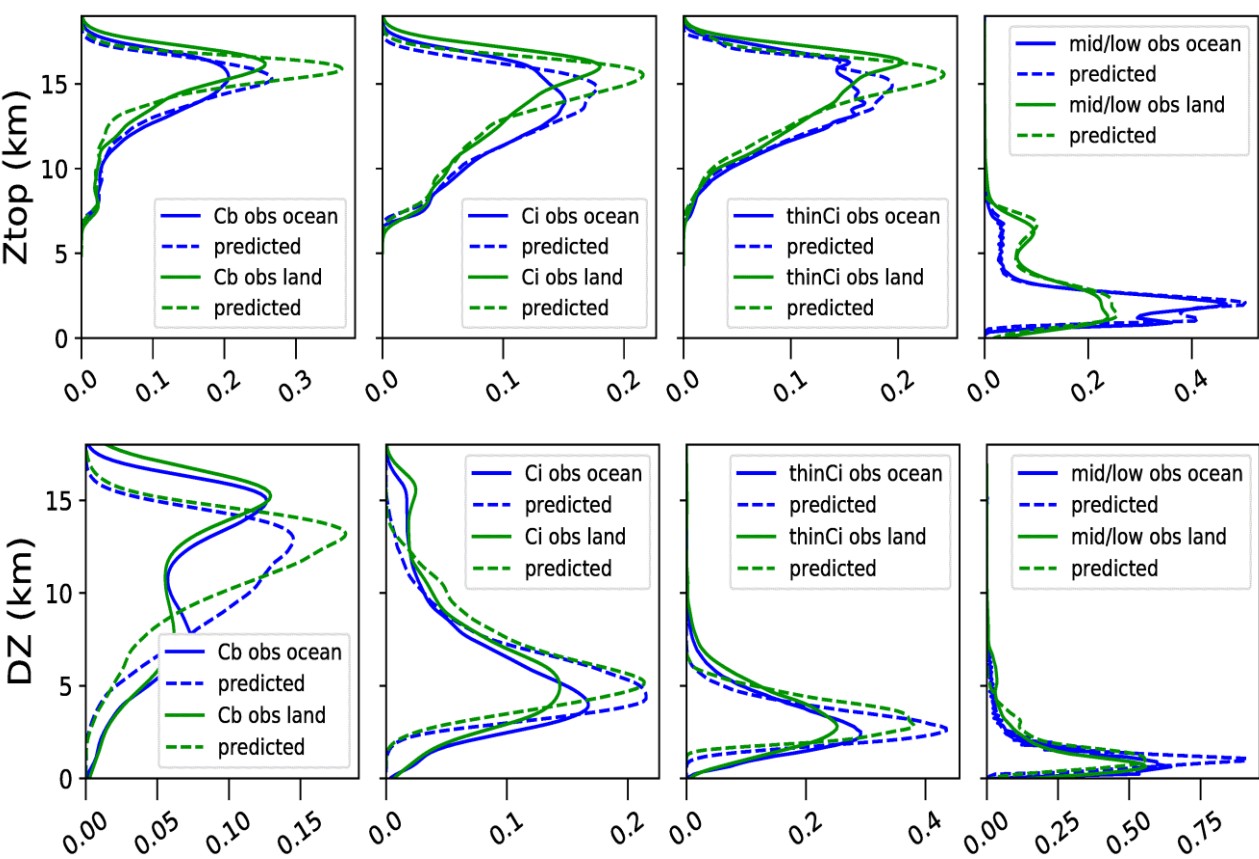

Figure 1: Normalized frequency distributions of Ztop (above) and DZ (below), separately for Cb, Ci, thin Ci and mid- / lowlevel clouds (identified by CIRS) and over ocean and land. The prediction models have been applied to 20% of the collocated data and are compared with the results derived from CloudSat-lidar 2B GEOPROF data.





**3 January 2008 (La Niña) 1:30AM / 9:30PM**   **18 January 2016 (El Niño) 1:30AM / 9:30PM**

**Cb      Ci      thin Ci    midlevel      lowlevel    clear sky**

0.5   2.5   4.5   6.5   8.5   10.5   12.5   14.5   16.5

0.5   2.5   4.5   6.5   8.5   10.5   12.5   14.5

0      0.5     1     1.5     2     2.5     3     3.5     4

**clouds above   clouds below  clouds above & below  UT clouds**

**Figure 2: Horizontal structure for one specific day during a La Niña (left) and El Niño situation (right) at 1:30AM and at 9:30PM LT of a) CIRS scene type b) $z_{top}$ (km), c) DZ (km), d) rain rate indicator and e) cloud layering of UT clouds.**





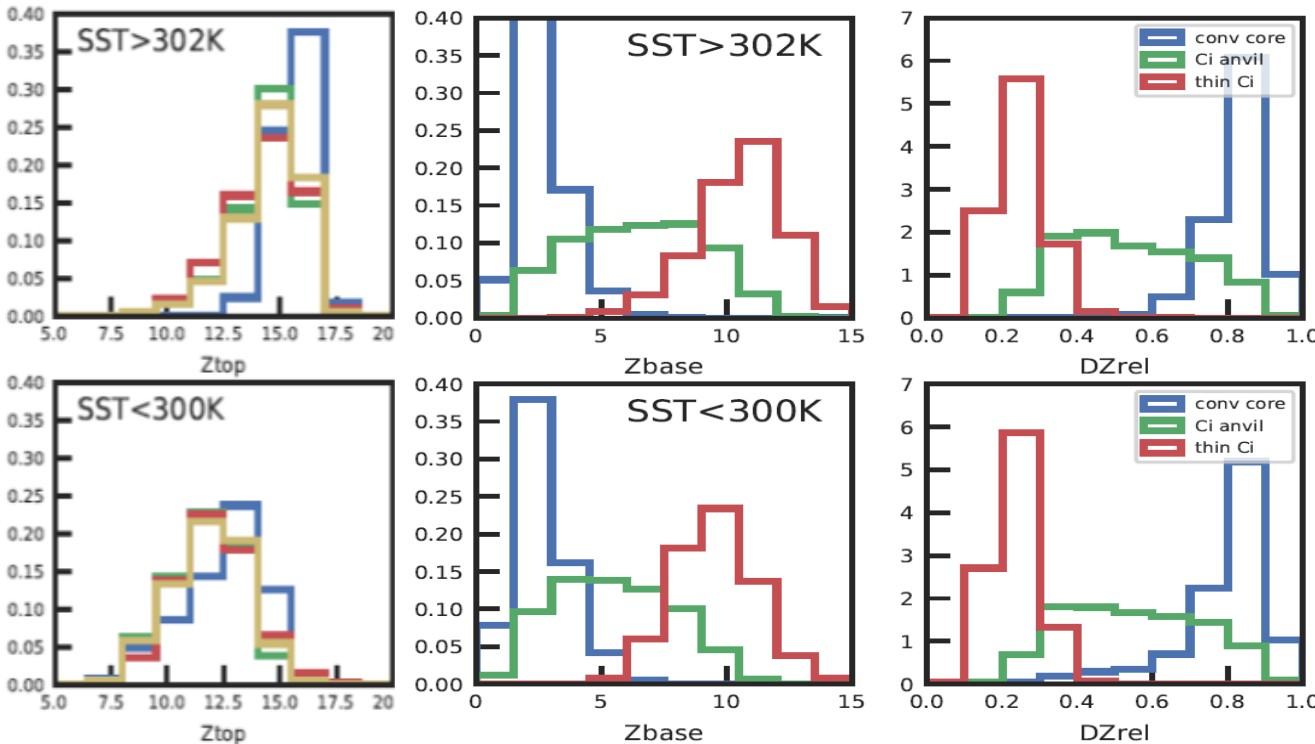

**Figure 3: Normalized distributions of cloud top height, cloud base height and relative vertical extent (DZ/Ztop) of MCS convective cores, Ci anvils and surrounding thin Ci. Statistics is for 2008-2018 at 1:30AM and 1:30PM, and distributions are compared over the 30% warmest regions (top) and the 30% coolest regions (bottom) over ocean.**





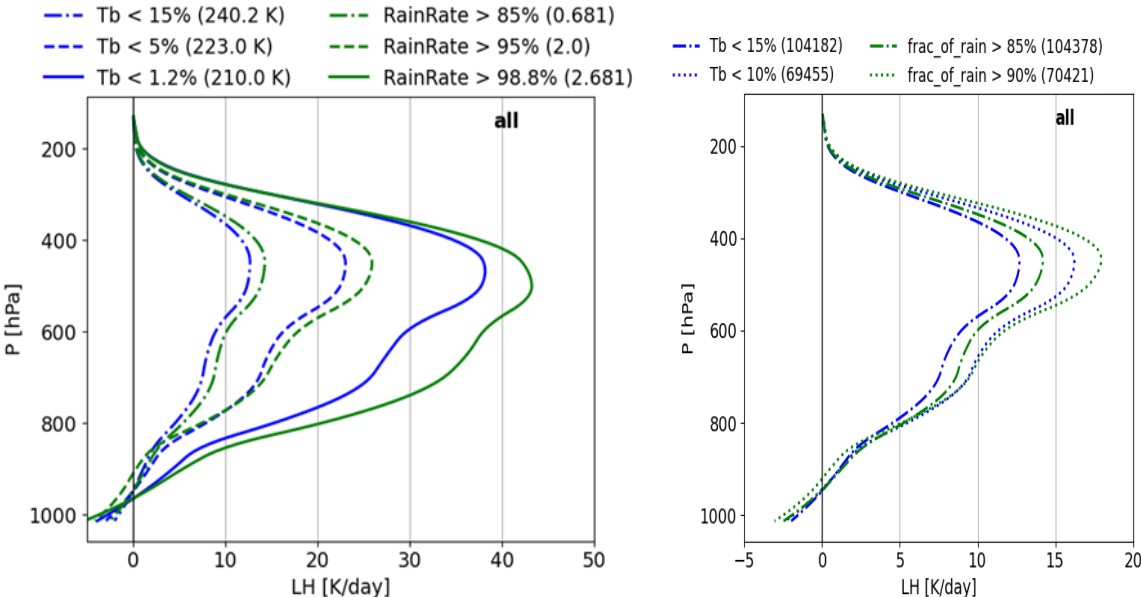

**Figure 4: Comparison of latent heating rate profiles averaged over the same percentile statistics, using the coldest brightness temperature (Tb), the largest rain rate indicator (right) and the largest spatial extension of rain within a grid cell of 0.5° (left). Since the grid cell precipitation coverage saturates at 1, one can only go down to the 10% largest cover. Statistics of collocated TRMM - AIRS data in the period 2008-2013.**

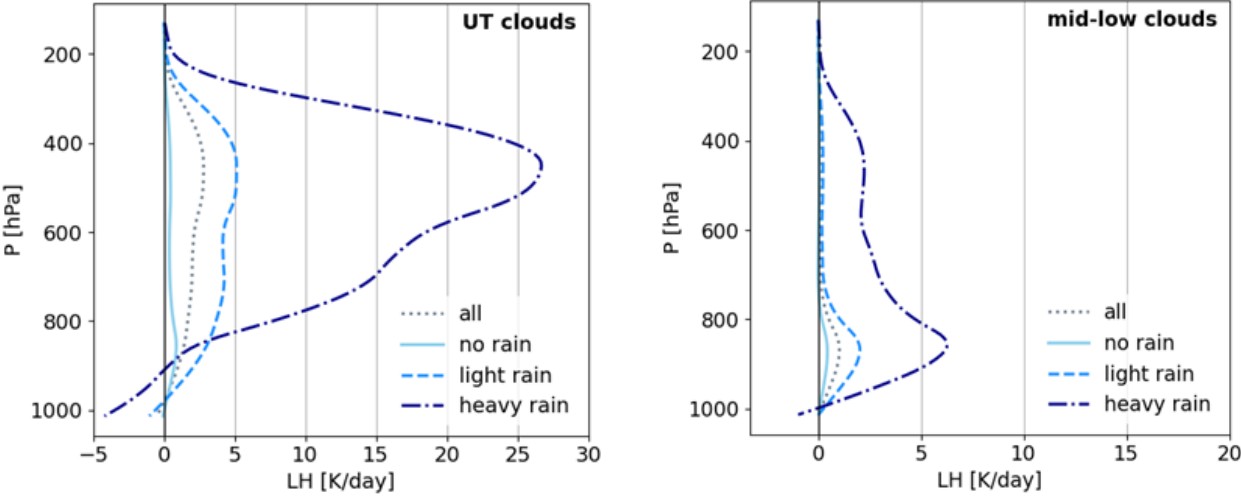

**Figure 5: Average latent heating profiles derived from TRMM (Shige et al., 2009) produced by UT clouds (left) and by mid- and low-level clouds (right) identified by AIRS, at the local time of 1:30 AM. In addition to the tropical mean are shown means of non-precipitating, lightly precipitating and heavily precipitating clouds. These precipitation conditions are given by the rain rate indicator classification derived from CloudSat and AIRS-ERA Interim machine learning. Statistics of collocated TRMM-AIRS in the period 2008-2013.**





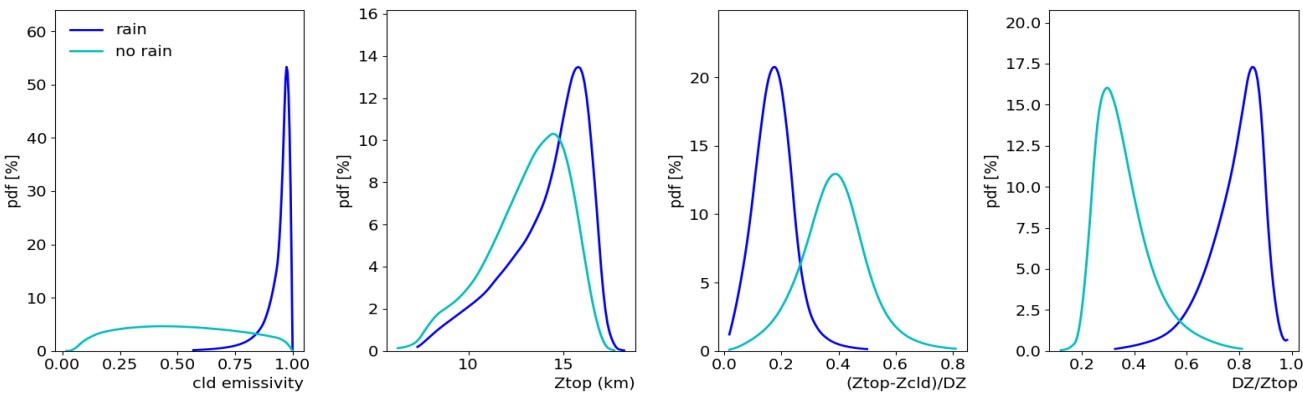

**Figure 6: Normalized distributions of cloud emissivity, cloud top height, difference between cloud top and CIRS near-top height, scaled by cloud vertical extent, and cloud vertical extent, scaled by cloud top height, for precipitating and non-precipitating UT clouds. Statistics is for 2008-2015 at 1:30AM.**


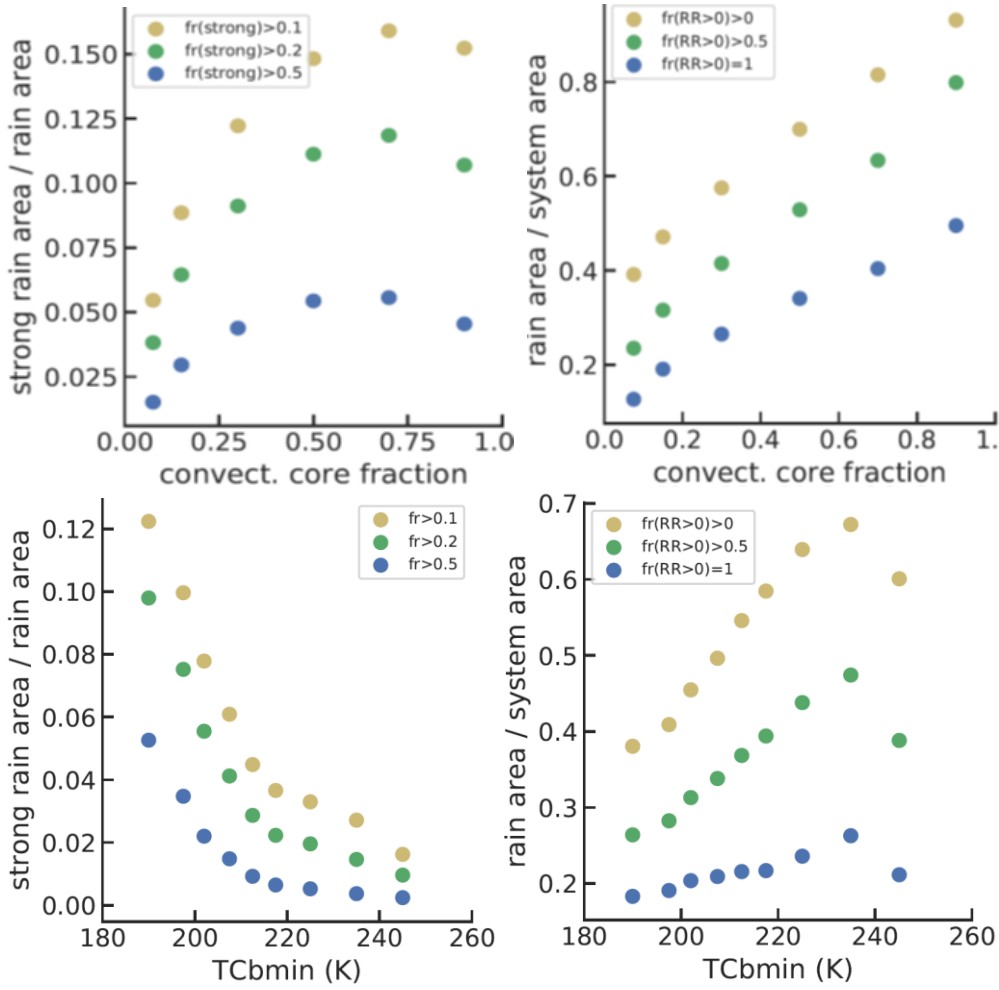

**Figure 7:** Ratio of relative strong rain area over precipitating area (left) and fraction of rainy area (right) within single core convective systems as a function of life cycle stage (top), given by decreasing convective core fraction, and within mature convective systems as a function of convective depth (bottom), given by decreasing minimum temperature within the convective cores. The convective cores are defined by cloud emissivity > 0.98, fraction of Cb within grid cell > 0.2 and $DZ/Z_{top}$ > 0.6. Different thresholds on the rain fraction per grid cell to be included into the areas are compared. The condition on strong rain also includes the condition that at least 50% of the grid cells are covered by any rain. Statistics combines observations at 1:30AM and 1:30PM, for 2008-2018.





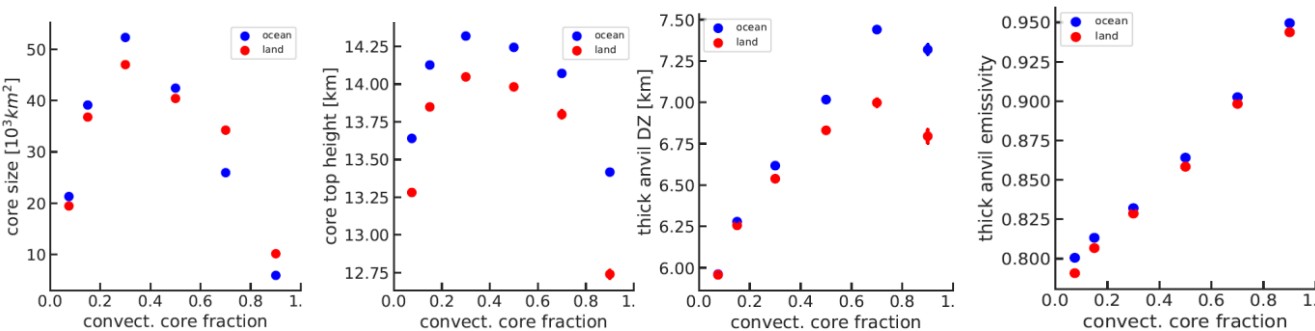

**Figure 8: MCS properties as function of their life cycle stage, given by fraction of convective core area within the system (1 corresponds to developing phase, with no anvil and 0.1 to dissipating stage): size and top height of the convective cores, vertical extent and emissivity of the thick anvils (emissivity > 0.5). Statistics combines observations at 1:30AM and 1:30PM, for 2008-2018.**

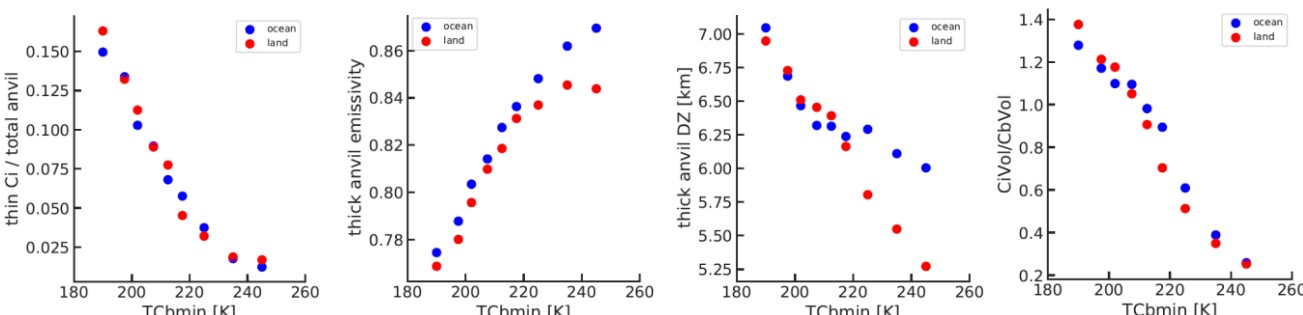

**Figure 9: Mature MCS system properties as function of their convective depth, given by the minimum temperature within the convective cores: ratio of areas with thin cirrus over total anvil, thick anvil emissivity and ratio of strong rain area over rainy area (right) within mature convective systems as a function of convective depth, given by decreasing minimum temperature within the convective cores. The convective cores are defined by cloud emissivity > 0.93, fraction of Cb within grid cell > 0.2 and DZ/Z$_{top}$ > 0.6. Different thresholds on the fraction of rain per grid cell to be included into the areas are compared. The condition on strong rain also includes the condition that at least 50 of the grid cells are covered by any rain. Statistics combines observations at 1:30AM and 1:30PM, for 2008-2018.**

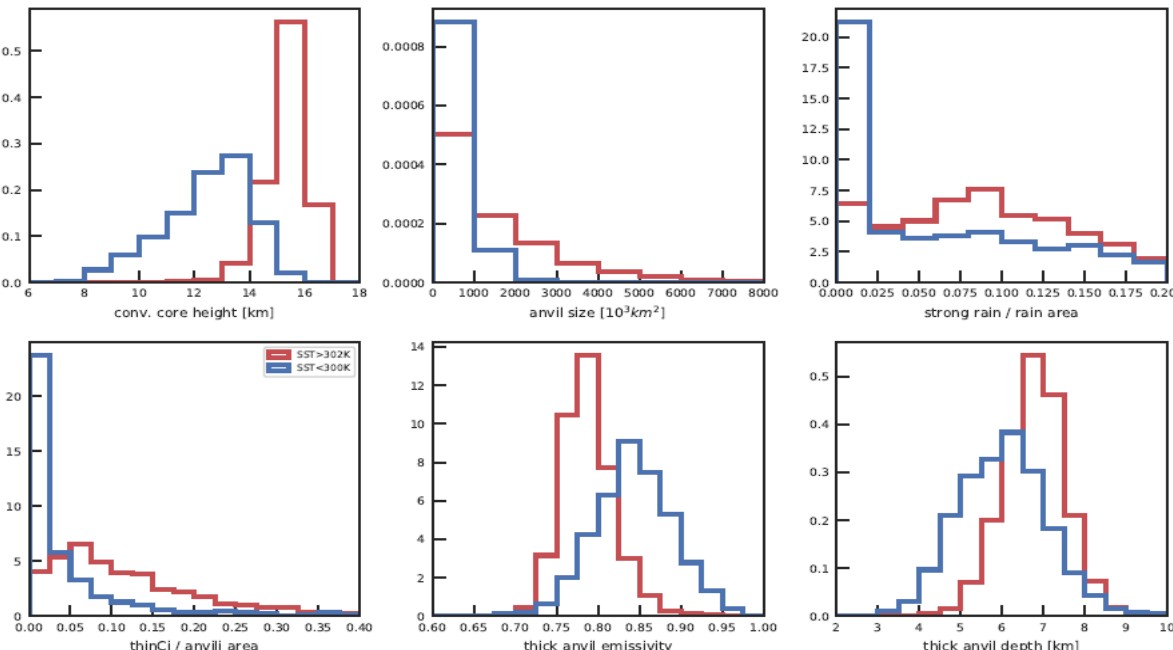

**Figure 10: Normalized frequency distributions of mature MCS system properties over the 30% warmest and coolest tropical ocean regions. Top panel rom left to right: height of convective cores, size of anvils, and ratio of areas with strong over total precipitation; bottom panel from left to right: ratio of areas with thin cirrus over total anvil, thick anvil emissivity, and thick anvil vertical extent. Statistics combines observations at 1:30AM and 1:30PM, for 2008-2018.**

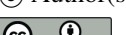



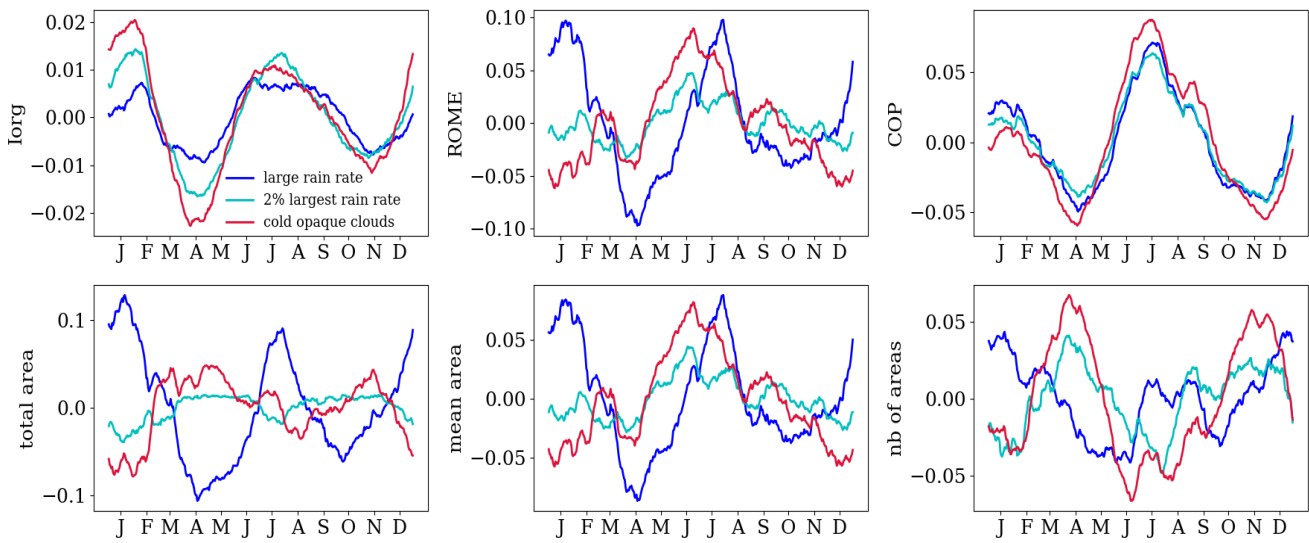

**Figure 11: Relative annual cycle of I$_{org}$, ROME, COP, total and mean convective areas and number of convective areas. These areas are built from grid cells covered by at least 90% UT clouds, with rain rate indicator > 2 (dark blue), Tcld < 230 K (red) and ε$_{cld}$ > 0.95, or using the 2% largest rain rate indicator (cyan). The latter leads to a constant total area of convection. Monthly**

**statistics of UT clouds averaged over four observation times from 2008 to 2018.**

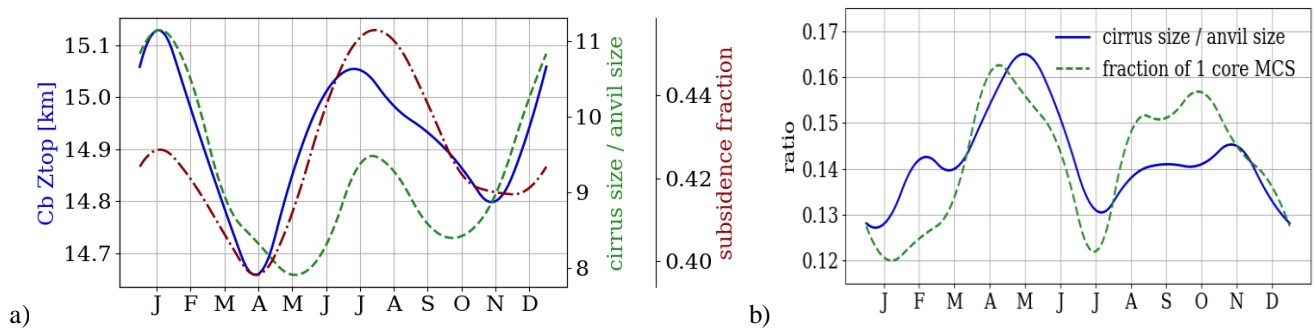

**Figure 12: right: Annual cycle of a) MCS core top height (blue), total anvil area over convective core area (red) and fraction of subsidence area (given by clear sky and low-level clouds) over the tropics (green) and b) percentage of thin cirrus over total anvil size (cyan) and of single core MCSs (red). Monthly statistics averaged over four observation times from 2008 to 2018.**





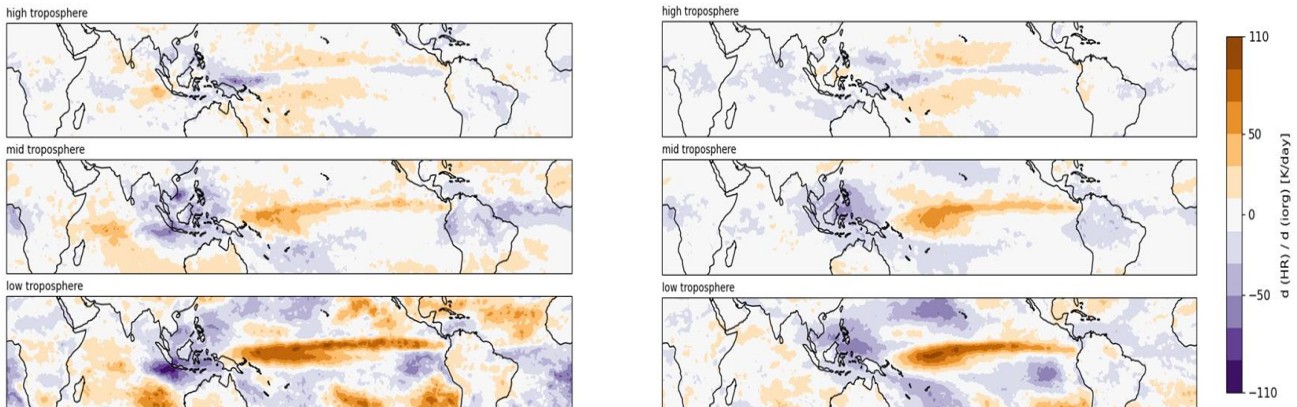

**Figure 13: Change in radiative heating rates with respect to deseasonalized $I_{org}$ computed from convective areas defined by grid cells with rain indicator > 2 (left) and by grid cells with $T_{cld} < 230$ K and $\varepsilon_{cld} > 0.95$ (right). The troposphere is divided into three layers: upper troposphere (100-200 hPa), mid troposphere (200-600 hPa), and low troposphere (600-900 hPa). Monthly statistics from 2008 to 2018.**

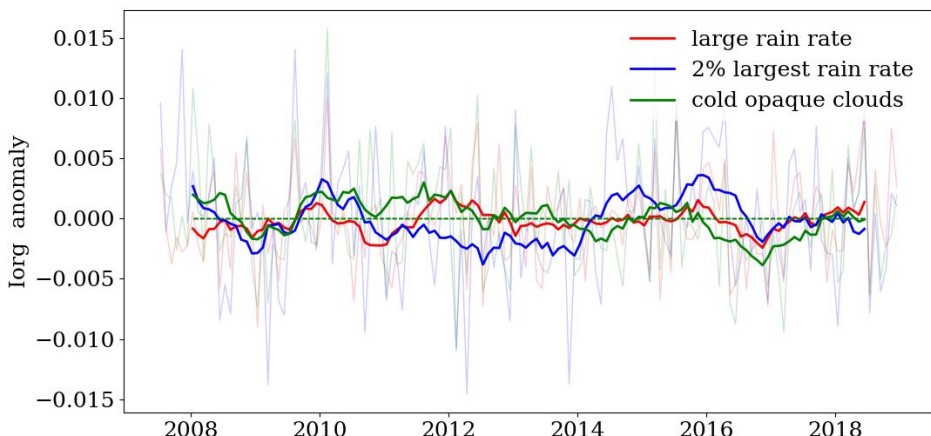

**Figure 14: Time series of deseasonalized monthly anomalies of $I_{org}$, using different proxies to define the convective areas. The deseasonalization was done by computing 12-month running means. The monthly anomalies are shown in light grey.**