# Peer review of "Convective Organization and 3D Structure of Tropical Cloud Systems deduced from Synergistic A-Train Observations and Machine Learning"

_Atmospheric Chemistry and Physics, 2022_

## Referee Comment (RC1)

**Review of "Convective Organization and 3D Structure of Tropical Cloud Systems deduced from Synergistic A-Train Observations and Machine Learning" by C. Stubenrauch et al.**

Atmospheric Chemistry and Physics

December 8, 2022

Stubenrauch et al. apply a machine-learning method to various satellite datasets to build a complete 3D description of upper tropospheric clouds. Their method fills gaps in the observational record due to sampling limitations. This allows the authors to examine metrics that are more closely related to physical processes than one can obtain with the raw satellite data alone. The technical analysis is very well done. I applaud the authors for carefully implementing the machine learning method and documenting how different necessary but subjective choices, such as thresholds and variable definitions, affect the results. However, the section that presents scientific results lacks coherence, and at times it reads like a list of findings that do not always connect to one another in an obvious way. This makes the writing unclear. I think the authors could improve the presentation of the results if they decide on a clear purpose for the paper and then write a more coherent presentation that fits that purpose. This could require some restructuring of the paper, so I recommend *major revision* for the paper. I would be happy to recommend this work for publication if my comments are addressed.

**General Comments**

- Section 3 presents a list of findings, first about relationships between anvil-cloud properties and deep convection, then proxies for the life cycle of mesoscale convective systems (MCS), then differences between MCS over warm sea-surface temperature and MCS over cool sea-surface temperature, then convective aggregation across the entire tropical belt. I was expecting a coherent link from one set of findings to another, and I didn't see this in the paper. I think the paper would be much clearer if the results were presented in a logical order in which each piece of analysis builds on the one before it, or if the authors explicitly state that they are investigating aspects of tropical convection that are related but do not necessarily build on one another as they are presented in the text. To accomplish this, it would help if the authors first decide the purpose of the paper. Is the purpose to document the machine-learning dataset and present some preliminary findings that can potentially be used for process-oriented evaluations of numerical models? (This is emphasized in the introduction as a key motivating factor.) If so, then please state in each subsection of section 3 that you are documenting distinct aspects of tropical convective clouds for potential use in model evaluations, and that the findings from one subsection do not directly build on the subsection before. Alternatively, is the purpose of the

paper to show new evidence about convective aggregation and climate? About half of the abstract text is devoted to this topic, but no clear conclusions are reached about this because the results are very sensitive to the particular aggregation index that is used. Furthermore, if the authors want to emphasize the convective aggregation results, then why use so much discussion on process-oriented behavior in Section 3.2 without explaining the link between these results and convective aggregation in Section 3.3? Overall, I could not identify the key messages that the authors want their readers to take away from the paper, other than the fact that they have produced a new dataset. This lack of a message is confusing. I recommend that the authors decide the purpose of the paper and the most important things that they want readers to learn, then revise the manuscript accordingly.

- It would help to provide a table that states the meaning of all variables and acronyms. It was hard for me to keep track of everything as I was reading, so I had to frequently stop and look up the meaning of different variables. I think it would help the reader if the authors provide a table with all of the definitions in one place.

**Specific Comments and Typos**

- Line 14: change "allows" to "allows us"
- Line 39: change "Pendergrast et al." to "Pendergrass" (only one author)
- Line 57: perhaps change "consolidating the hypothesis" to "supporting the hypothesis"
- Line 71: change "become" to "have become"
- Line 122: change "skewedness" to "skewness"
- Line 132: perhaps change "we use randomly chosen 80% of the dataset" to "we use 80% of the dataset chosen at random"
- Line 153: it would help to define DZ in the text here. DZ is defined in Table 1, but by the time I read this sentence I did not remember what DZ stands for, so I had to pause and look it up. It would be easier for the reader if you define DZ here because this is the first place where it is referenced in the text.
- Line 200: remove commas after "both" and "classification"
- Line 238: change "as convective core" to "as a convective core"
- Line 240: remove comma after "UT clouds"
- Line 314: change "area of the larger area" to "area of the larger object"?
- Line 318: "we highlight results on the ML derived variables by investigation relationships". Relationships between what? Please specify.
- Line 328: change "grid cells which include heavy precipitation" to "grid cells that include heavy precipitation"
- Line 416: change "the less sensitive" to "the least sensitive"
- Line 436: change "Iorg is the metric that is less related to these variables" to "Iorg is the metric that is least related to these variables"

- Line 547: the README page for the AIRS data from the CIRS L2 webpage is not working. This makes it difficult for others to obtain and use the data. Are the authors able to fix this?
- Figure 1: the legend of the bottom left panel partly covers the curves in the plot. This is confusing because the text mentions that the curves have a bimodal distribution (line 173), but one of the modes of the distribution is covered by the legend in Fig. 1. Please revise Fig. 1 so that all curves are not obscured by the legends.
- Figure 9: Only three variables are listed in the caption ("Mature MCS system properties as function of their convective depth, given by the minimum temperature within the 735 convective cores: ratio of areas with thin cirrus over total anvil, thick anvil emissivity and ratio of strong rain area over rainy area (right) within mature convective systems as a function of convective depth, given by decreasing minimum temperature within the convective cores.") However, four subplots are shown, so some description is missing. Please consider adding labels (a), (b), (c), etc. to the subplots and including these labels in the caption. This will clarify the figure because the variable names on the vertical axes are not stated in the caption.
- Figure 13: Change "high troposphere" to "upper troposphere" on the figure for consistency with the caption. Alternatively, you could state the pressure ranges on the figure so the reader can see the levels of each subplot without looking back and forth at the caption (e.g. replace "high troposphere" with "100-200 hPa" on the figure, and so on).

---

## Author Comment (AC1)

**Author response to Referees 1 and 2:**

Review of "Convective Organization and 3D Structure of Tropical Cloud Systems deduced from Synergistic A-Train Observations and Machine Learning" by C. Stubenrauch et al.

**General Comments of Referee 1:**

• Section 3 presents a list of findings, first about relationships between anvil-cloud properties and deep convection, then proxies for the life cycle of mesoscale convective systems (MCS), then differences between MCS over warm seasurface temperature and MCS over cool sea-surface temperature, then convective aggregation across the entire tropical belt. I was expecting a coherent link from one set of findings to another, and I didn't see this in the paper. I think the paper would be much clearer if the results were presented in a logical order in which each piece of analysis builds on the one before it, or if the authors explicitly state that they are investigating aspects of tropical convection that are related but do not necessarily build on one another as they are presented in the text. To accomplish this, it would help if the authors first decide the purpose of the paper. Is the purpose to document the machine-learning dataset and present some preliminary findings that can potentially be used for process-oriented evaluations of numerical models? (This is emphasized in the introduction as a key motivating factor.) If so, then please state in each subsection of section 3 that you are documenting distinct aspects of tropical convective clouds for potential use in model evaluations, and that the findings from one subsection do not directly build on the subsection before. Alternatively, is the purpose of the paper to show new evidence about convective aggregation and climate? About half of the abstract text is devoted to this topic, but no clear conclusions are reached about this because the results are very sensitive to the particular aggregation index that is used. Furthermore, if the authors want to emphasize the convective aggregation results, then why use so much discussion on process-oriented behavior in Section 3.2 without explaining the link between these results and convective aggregation in Section 3.3? Overall, I could not identify the key messages that the authors want their readers to take away from the paper, other than the fact that they have produced a new dataset. This lack of a message is confusing. I recommend that the authors decide the purpose of the paper and the most important things that they want readers to learn, then revise the manuscript accordingly.

**and of Referee 2:**

This study aims to study convective organization and dynamics of upper tropospheric clouds over the global tropics using satellite datasets and ERA reanalysis products. The authors apply novel machine learning method to merge these datasets in order to fill in the gap in these dataset to obtain a fully-connected understanding of tropical convective systems. Overall, I commend the authors for writing such a comprehensive manuscript and I really liked the scientific analyses presented here. I specifically liked the authors efforts to connect dots between convection, cloud systems, and organization using multiple metrics so as to provide a better idea of tropical convection. However, I think that manuscript needs some restructuring as currently it looks like a lot of ingredients are mixed together but it didn't result in an edible dish. I got lost and a bit confused in section 3 as the authors jump between multiple thoughts and I couldn't connect the dots well. Therefore, I recommend **major revision** to the manuscript.

**The authors thank both reviewers for their very constructive comments. They both agree that the description of the results and conclusions and abstract miss clarity. Indeed, since this dataset was also new for the authors, they wanted to show as many results as possible, but this made the manuscript difficult to follow and did not give clear key messages. Therefore the authors have completely revised section 3 (Results), have added a clear goal of this article into the introduction and have shortened and clarified the abstract and section 4 (conclusions and Outlook), so that the key messages come out clearer. In the following we copied the changes made to the manuscript into this document.**

**- We have added in the introduction just after the first paragraph the goal:**

*The goal of this article is to present a coherent long-term 3D dataset which describes tropical UT cloud systems and which can be used on one hand for a process-oriented evaluation of convective parameterizations in climate models and on the other hand for the study of convective organization.*

**- We have also reformulated the last paragraph of the introduction:**

*Apart from the conclusions and outlook given in Section 4, the article is divided into two main sections: Section 2 describes the data, methods and evaluation and Section 3 highlights scientific results which show the applicability of these newly derived variables.*

*Section 2 first describes the collocated data, the neural network development as well as an evaluation of the predictions on the collocated data. In addition, it presents the creation of the 3D dataset containing the additional variables (2.3) and the cloud system reconstruction (2.4). The last subsection (2.5) gives a short overview of existing convective organization indices and proxies for defining the convective objects. Section 3 first shows the coherence of these ML-derived properties, in particular the rain intensity classification, using the complete 3D dataset (section 3.1). Then, in combination with a cloud system analysis, section 3.2 presents the MCS properties with respect to their life cycle stage and their convective depth. The last subsection (3.3) explores tropical convective organization: we compare different proxies for convection and resulting indices of convective organization, by investigating annual cycle and inter-annual variability. The latter is small over the considered time period (2008 – 2018), but we find interesting geographical patterns in changes of radiative heating rate fields in relation to the tropical convective organization.*

**- We have also shortened the last part of the abstract:**

*… This rain intensity classification is more efficient to detect large latent heating than cold cloud temperature. In combination with a cloud system analysis we found that deeper convection leads to larger heavy rain areas and a larger detrainment, with a slightly smaller thick anvil emissivity. This kind of analysis can be used for a process-oriented evaluation of convective precipitation parameterizations in climate models. Furthermore we have shown the usefulness of our data to investigate tropical convective organization metrics. A comparison of different tropical convective organization indices and proxies to define convective areas has revealed that all indices show a similar annual cycle in convective organization, in phase with convective core height and anvil detrainment. The geographical patterns and magnitudes in radiative heating rate inter-annual changes with respect to one specific convective organization index ($I_{org}$) for the period 2008 to 2018 are similar to the ones related to the El Niño Southern Oscillation. However, since the inter-annual anomalies of the convective organization indices are very small and noisy, it was impossible to find a coherent relationship with those of other tropical mean variables such as surface temperature, thin cirrus area or subsidence area.*

**- We have rewritten section 3 (results), with a short introduction about the structure of this section:**
*3. Results*

*As application examples we highlight results from analyses using this long-term 3D dataset. We particularly concentrate our interest on the ML-derived rain rate indicator. Section 3.1 shows the coherence of this newly derived variable. The cloud system approach enables us to study the behaviour of the MCSs with respect to their life cycle stage and convective depth. This process-oriented analysis presented in section 3.2 can be used to evaluate parameterizations in climate models (Stubenrauch et al., 2019). In section 3.3, we show results concerning mesoscale convective organization. Mesoscale convective organization has been identified by larger and higher systems, which also live longer than unorganized systems (e. g. Rossow and Pearl, 2007; Takahashi et al., 2021), and they also lead to increases in tropical rainfall (e. g. Tan et al., 2015). We first compare convective organization indices derived from objects defined by strong rain and by cold cloud temperature and then investigate changes in geographical patterns of radiative heating with respect to one of these indices ($I_{org}$).*

**- We have changed the subtitle of section 3.1 from 'Properties of UT clouds' to:**
*3.1 Coherence of ML-derived rain intensity classification*

**- We have taken out the titles of the two subsections ('Behaviour of precipitating areas' and 'Behaviour of convective core and anvil properties') of section 3.2 and have really focused on the 'process-oriented behaviour of mesoscale convective systems' for the evaluation of climate models, with merged figures, so that this subsection only includes two figures instead of four. We hope that section 3.2 is much clearer now:**
*3.2 Process-oriented behaviour of mesoscale convective systems*

[revised manuscript text omitted]

**- Section 3.3 is indeed not linked to section 3.2, but shows just another example of application of our new 3D dataset. We have stated this in the introduction and in the paragraph following the title of section 3 (see above).**

**- We have also reformulated the first part of section 3.3 (comparison of convective organization indices by means of their annual cycle) to make it clearer for key messages:**

[revised manuscript text omitted]

**- We have also added a figure with histograms at the end of section 3.3 (analysis of inter-annual variablity), again for a clearer key message:**

*…*

*Whereas the geographical patterns of the derivatives of heating / cooling with respect to $I_{org}$ show a coherent picture, we did not find any correlation between the very small inter-annual anomalies of $I_{org}$ (shown in Figure 12) and the ones of the tropical means of different variables like surface temperature, thin cirrus area and subsidence area. The correlations depend on the proxies for the definition of the convective areas and in particular on the metrics for convective organization. Already the time series of the inter-annual anomalies of the different indices have a different behaviour as can be seen in Figure S8 in the supplement. We have also investigated tighter thresholds on the variables which define deep convection (like rain rate indicator > 2.5 or $T_{cld}$ < 210 K), however we are left with only about 0.5% total area, which increases the noise level. In addition, we found that the results also change when we exclude objects with the size of only one grid cell (not shown), as already pointed out by Jin et al. (2022). Therefore we do not consider it meaningful to use the discussed convective organization indices for an estimation of tropical mean changes with respect to changes in convective organization.*

*While we have seen that the convective organization indices vary much more seasonally than inter-annually, Figure 13 suggests that the difference of the density distributions of convective core height*

*and strong rain area within the MCSs between April and July or between cool years (2008/11) and warm years (2015/16) is of the same order, with a shift towards higher core height and a longer tail in strong rain area. However, the size distributions of the MCSs are similar. The tail in the mean area of strong precipitation within the MCSs is clearly larger in the case of warmer years. This indicates that a shift in tropical surface temperature changes only a small part of the MCSs, with more extreme values. Such a behavior cannot be identified using a convective organization index computed over the whole tropics.*

[Figure]

Figure 12: Time series of deseasonalized monthly anomalies of I$_{org}$, using different proxies to define the convective areas. The deseasonalization was done by computing 12-month running means. The monthly anomalies are shown in light grey.

[Figure]

Figure 13: Density distributions of MCS system properties, comparing April and July (top) and cooler and warmer years (bottom): height of convective cores (left), size of areas with strong precipitation (middle) and system size (right). Statistics combines observations at 1:30AM and 1:30PM, for 2008-2018, for MCSs with core fraction > 0.1.

**- We have shortened and rewritten parts of section 4, so that key messages come out clearer, which are pointed out in italic:**

**4. Conclusions and Outlook**

We have presented a methodology to extend spatially and temporally information on the cloud vertical structure and precipitation derived from active lidar and radar measurements of CALIPSO and CloudSat missions. This new approach made use of CIRS data obtained from advanced IR sounder measurements of AIRS and IASI combined with ERA-Interim reanalyses and machine learning technologies using ANN. *The resulting 3D dataset of UT cloud systems, covering 2008 to 2018, together with a similarly produced dataset of radiative heating rates (Stubenrauch et al., 2021), can be used to improve our understanding*

*of the relationship between tropical convection and resulting anvils and how they are impacted by and feed back to climate change.*

Though the uncertainties in the predicted variables and classifications are relative large (with an accuracy of about 65 to 70% for the rain intensity classification), this new dataset allows to study their *horizontal structures on specific snapshots in time*. For a complete instantaneous coverage, necessary to compute indices of tropical convective organization, the gaps between the orbits have been filled iteratively with the four observations per day of AIRS and IASI data, starting with those closest in time (already leading to 90% coverage). We have demonstrated that *the newly developed precipitation intensity classification is slightly more efficient to detect large latent heating and therefore deep convection compared to the cold cloud temperature.*

The cloud system approach developed by Protopapadaki et al. (2017) has been slightly modified, and the normalized vertical extent obtained from the ML approach has been employed to slightly improve the identification of the convective cores, in particular in the cooler tropical regions. *The cloud system concept allows a process-oriented evaluation of parameterizations in climate models. In agreement with earlier studies (e. g. Schumacher and Houze, 2003; Roca et al., 2014, Takahashi et al., 2021), we found that deeper convection leads to larger areas of heavy rain. These results also confirm the quality of the ML-derived precipitation rate classification. With increasing convective depth mature MCSs also show an increase in volume detrainment, while the anvil emissivity slightly decreases.*

Moreover we have shown the *usefulness of our new dataset by investigating convective organization metrics*. By comparing different organization indices ($I_{org}$, COP, ABCOP and ROME) and proxies to define convective objects, we have shown that the indices indicate a similar annual cycle of convective organization. *However, ABCOP and ROME are strongly correlated to the total and mean area of the objects, respectively. While the mean area of the objects is certainly an indication of convective organization, their total area at tropical scale seems to be less linked to organization.* The index $I_{org}$, which only considers the distance between convective objects, seems to add another information. The core height of the MCSs and their anvil detrainment are in phase with the annual cycle of $I_{org}$ and COP, as well as the relative subsidence area. This shows also a link between the MCSs and subsidence areas. It is interesting to note that the annual cycles of the total area of cold cloud objects and of intense precipitation objects are very different. This can be related to a nearly opposite cycle in their mean size and number for the first, and to a cycle in phase for the latter.

Changes in gradients of tropospheric radiative heating relate to changes in atmospheric circulation. The geographical patterns and magnitudes in radiative heating rate changes with respect to $I_{org}$ are similar for both proxies, but slightly larger for strong precipitation areas. This may be expected as *intense precipitation should be a more direct proxy for convection than cold cloud top. Furthermore the HR pattern changes are similar to the ones related to ENSO during this period.*

However, the *time series of the inter-annual anomalies of convective organization strongly depend on the convective organization metrics, and correlations between these anomalies and those of tropical means of different atmospheric variables do not show consistent results*. The tail of the distribution of strong rain areas seems to be more related to warmer tropics than the indices themselves. Therefore *one has to be careful using only one of these organization indices and proxies to study climate change*. More detailed studies are necessary to show the behaviour of these indices with spatial resolution and domain size.

This data base of UT cloud systems, their vertical structure and precipitation areas is being constructed within the framework of the GEWEX (Global Energy and Water Exchanges) Process Evaluation Study on Upper Tropospheric Clouds and Convection (GEWEX UTCC PROES) to advance our knowledge on the climate feedbacks of UT clouds. It will be made available within this year via https://gewex-utcc-proes.aeris-data.fr/. For the future it will also be interesting to use this dataset for the study of cold pools, using data of Garg et al. (2020).

In order to continue this dataset beyond 2018, we are now preparing a new version of CIRS data, using ERA5 (Hersbach et al., 2020) instead of ERA-Interim ancillary data, and newly calibrated AIRS L1C radiances (Manning et al., 2019) as input.

**Specific comments of Referee 1:**

• It would help to provide a table that states the meaning of all variables and acronyms. It was hard for me to keep track of everything as I was reading, so I had to frequently stop and look up the meaning of different variables. I think it would help the reader if the authors provide a table with all of the definitions in one place.

**- As some variables (in particular 'cloud fuzziness') have been only explained in section 3, which made it more difficult to follow the discussion of the results, we made sure that all variables are explained and named in section 2.1; the variables derived from the target variables are now also listed, together with their computation from, in Table 1. In the discussion of the results we now use the variables name instead of the formulas.**

**Specific Comments and Typos**
• Line 14: change "allows" to "allows us"
• Line 39: change "Pendergrast et al." to "Pendergrass" (only one author)
• Line 57: perhaps change "consolidating the hypothesis" to "supporting the hypothesis"
• Line 71: change "become" to "have become"
• Line 122: change "skewedness" to "skewness"
• Line 132: perhaps change "we use randomly chosen 80% of the dataset" to "we use 80% of the dataset chosen at random"
*All taken into account*

• Line 153: it would help to define DZ in the text here. DZ is defined in Table 1, but by the time I read this sentence I did not remember what DZ stands for, so I had to pause and look it up. It would be easier for the reader if you define DZ here because this is the first place where it is referenced in the text.
*Done, and we also write in section 3 'vertical extent', so that the discussion is easier to follow*

• Line 200: remove commas after "both" and "classification"
• Line 238: change "as convective core" to "as a convective core"
• Line 240: remove comma after "UT clouds"
• Line 314: change "area of the larger area" to "area of the larger object"?
*All taken into account*

• Line 318: "we highlight results on the ML derived variables by investigation relationships". Relationships between what? Please specify.
*We changed this sentence to:*
*As application examples we highlight results from analyses using this long-term 3D dataset.*

• Line 328: change "grid cells which include heavy precipitation" to "grid cells that include heavy precipitation"
• Line 416: change "the less sensitive" to "the least sensitive"
• Line 436: change "Iorg is the metric that is less related to these variables" to "Iorg is the metric that is least related to these variables"
*All taken into account*

• Line 547: the README page for the AIRS data from the CIRS L2 webpage is not working. This makes it difficult for others to obtain and use the data. Are the authors able to fix this?

***Thank you very much for pointing this out! The first author has contacted the French Data Centre
AERIS, and they have repaired this*** *(this is unfortunately not the first time that this happened…)****, and
we have also put a readme for the available variables directly at https://cirs.aeris-data.fr/)***

• Figure 1: the legend of the bottom left panel partly covers the curves in the plot.
This is confusing because the text mentions that the curves have a bimodal distribution (line 173),
but one of the modes of the distribution is covered by the legend in Fig. 1. Please revise Fig. 1 so that
all curves are not obscured by the legends.
***Taken into account***

• Figure 9: Only three variables are listed in the caption ("Mature MCS system properties as function
of their convective depth, given by the minimum temperature within the 735 convective cores: ratio
of areas with thin cirrus over total anvil, thick anvil emissivity and ratio of strong rain area over rainy
area (right) within mature convective systems as a function of convective depth, given by decreasing
minimum temperature within the convective cores.") However, four subplots are shown, so some
description is missing. Please consider adding labels (a), (b), (c), etc. to the subplots and including
these labels in the caption.
This will clarify the figure because the variable names on the vertical axes are not stated in the
caption.
***Figure  7 has been removed as a separate figure and Figures 8 and 9 (now Figures 7 and 8) are new
(see above)***

• Figure 13: Change "high troposphere" to "upper troposphere" on the figure for consistency with
the caption. Alternatively, you could state the pressure ranges on the figure so the reader can see
the levels of each subplot without looking back and forth at the caption (e.g. replace "high
troposphere" with "100-200 hPa" on the figure, and so on).
***Taken into account***

**Specific comments of Referee 2:**
1) First, some figures are highly pixelated (Figures 3, 7, 10, 11) and therefore I recommend providing
high resloution version of all the figures.

***Taken into account; the author have had problems to include the figures in pdf format into the word
text, which reduced the initial resolution. All figures will be separately provided in good resolution
(This will be checked also by the editors before publishing)***

2) I was wondering that whether the authors performed any tests regarding the variables of interest?
For e.g., did the authors test other atmospheric state variables to predict the cloud properties? If not,
can the authors comment on how they came up with these input/output variables. It would also be
useful if the authors can comment on relative importance of each input variables, if possible. This will
aid the readers to choose which variables are more important and significant for cloud properties
and rain classification.

***The authors have done this kind of test for the radiative heating rates (published in Stubenrauch et
al., 2021, https://doi.org/10.5194/acp-21-1015-2021): one surprising result was that once basic
surface, atmospheric and cloud properties were included, the results were already very good, and by
adding other variables, the results improved only by 5 to 10% (section 3.1 of the article from 2021).
We have done similar tests for the variables published in this article, with a similar results. For the
choice of the input variables, we started with the CIRS cloud properties, and added then atmospheric
and surface properties from ERA-Interim. The latter have been used in the CIRS cloud property
retrieval, therefore the whole input variables are coherent. We found out that the scenes for which***

*the models were developed are much more important than the detailed choice of the input variables (once the basic variables were included). We also tested the resolution of the atmospheric profiles, by using once 10 and once 20 layers, but again there was not much difference in the results.*

3) I suggest adding a schematic of the designed ANN as it would be easier to visualize the connected network and their hidden layers.

*This ANN network approach is actually quite standard, so we do not want to add an additional figure on it. Just for your information, it would look a bit like this:*

[Figure]

4) Previous studies (de Szoeke et al. 2017, Garg et al., 2020) have shown that cold pools have a strong relationship with tropical convection and specially with precipitation and convective areas. I highly recommend connecting these results with global tropical oceanic cold pool properties shown in these studies as it will help the authors connect dots between clouds, precipipitation and convection over the global tropics.

*Thank you very much for these articles! Indeed, in the future we want to combine our 3D dataset with others, in order to understand better the different interactions. We have added a sentence in section 4, and the first author also shared this information within the GEWEX UTCC PROES group.*

*For the future it will also be interesting to use this dataset for the study of cold pools, using data of Garg et al. (2020).*

5) Line 390: Correct "MSCs" to "MCSs".

6) I suggest correcting TB to $T_B$ throughout the manuscript.

***Both taken into account***

7) I recommend summarizing the conclusions section in some take-away points so that it better concludes and summarize your major points. Currently the conclusions section is also not linked at all.

**We have shortened and rewritten parts of section 4, so that key messages come out clearer, which are pointed out in italic (see above).**

8) In the Code/Data availability, the authors have provided links to the relevant datasets but I am just wondering if the authors intend to provide open source code of their ANN as well?

*We will provide the 3D dataset (HR's, vertical structure and rain rate indicator) via the GEWEX UTCC PROES website. The first author has already contacted the French data centre AERIS and will start as soon as possible with the documentation, and this distribution should be within this year. In the meantime we continue to test the data. We do not foresee to distribute the ANN code, because they a several programs (training, data production and evaluation), they have not been written in such a way that they will be easy to use, and we do not have any funding for rewriting and document them.*

*We prefer to concentrate our effort on providing and document the dataset (which needs extensive testing).*